# An entropic safety catch controls hepatitis C virus entry and antibody resistance

Lenka Stejskal[1,2†], Mphatso D Kalemera[1†], Charlotte B Lewis[3], Machaela Palor[1], Lucas Walker[1], Tina Daviter[2,4], William D Lees[2], David S Moss[2], Myrto Kremyda-Vlachou[5], Zisis Kozlakidis[6], Giulia Gallo[7], Dalan Bailey[7], William Rosenberg[8], Christopher JR Illingworth[3,9,10,11], Adrian J Shepherd[2], Joe Grove[1,3]*

[1]Institute of Immunity and Transplantation, Division of Infection and Immunity, University College London, London, United Kingdom; [2]Institute of Structural and Molecular Biology, Birkbeck College, London, United Kingdom; [3]MRC-University of Glasgow Centre for Virus Research, Glasgow, United Kingdom; [4]Shared Research Facilities, The Institute of Cancer Research, London, United Kingdom; [5]Division of Infection and Immunity, University College London, London, United Kingdom; [6]International Agency for Research on Cancer, World Health Organization, Lyon, France; [7]The Pirbright Institute, Pirbright, United Kingdom; [8]Division of Medicine, Institute for Liver and Digestive Health, University College London, London, United Kingdom; [9]Department of Genetics, University of Cambridge, Cambridge, United Kingdom; [10]Institut für Biologische Physik, Universität zu Köln, Cologne, Germany; [11]MRC Biostatistics Unit, University of Cambridge, Cambridge, United Kingdom

*For correspondence:
Joe.Grove@glasgow.ac.uk

†These authors contributed equally to this work

**Abstract** E1 and E2 (E1E2), the fusion proteins of Hepatitis C Virus (HCV), are unlike that of any other virus yet described, and the detailed molecular mechanisms of HCV entry/fusion remain unknown. Hypervariable region-1 (HVR-1) of E2 is a putative intrinsically disordered protein tail. Here, we demonstrate that HVR-1 has an autoinhibitory function that suppresses the activity of E1E2 on free virions; this is dependent on its conformational entropy. Thus, HVR-1 is akin to a safety catch that prevents premature triggering of E1E2 activity. Crucially, this mechanism is turned off by host receptor interactions at the cell surface to allow entry. Mutations that reduce conformational entropy in HVR-1, or genetic deletion of HVR-1, turn off the safety catch to generate hyper-reactive HCV that exhibits enhanced virus entry but is thermally unstable and acutely sensitive to neutralising antibodies. Therefore, the HVR-1 safety catch controls the efficiency of virus entry and maintains resistance to neutralising antibodies. This discovery provides an explanation for the ability of HCV to persist in the face of continual immune assault and represents a novel regulatory mechanism that is likely to be found in other viral fusion machinery.

## Editor's evaluation

HCV is unique in its glycoprotein structure, complex receptor usage and an unusual persistence for a (+)RNA virus. In this well done study, the authors explain a number of observations regarding receptor usage and how HCV evades antibody control via HVR1, whose disordered nature enable mutation to continually evade antibody responses.

## Introduction

The fusion proteins of enveloped viruses are 'spring-loaded' for dramatic conformational changes that force viral and host membranes together. This requires careful regulation: the conformational switch is, generally, irreversible and premature triggering inactivates the machinery; conversely, failure to trigger will prevent successful entry (*Kielian and Rey, 2006*; *Rey and Lok, 2018*). Therefore, fusion proteins rely on molecular cues to ensure timely and effective activation. For viruses that establish chronic infections (HCV, HIV, HBV etc.) this is achieved in the face of neutralising antibodies (nAbs) disrupting the function and regulation of their entry proteins. A molecular understanding of virus entry has guided therapeutic and vaccine design, and provided insights into fusion mechanics in eukaryotes (*Burton et al., 2012*; *Crank et al., 2019*; *Fédry et al., 2017*; *McLellan et al., 2013*; *Vance and Lee, 2020*).

Viral fusion proteins are broadly categorized as class-I, II, or III fusion machines, with diverse viruses within each class exhibiting similar mechanisms. For example, HIV, Ebolavirus, and Coronaviruses all exhibit structurally similar class-I fusion machinery (*Rey and Lok, 2018*). However, the glycoproteins of Hepatitis C Virus (HCV), E1 and E2 (E1E2), do not possess the hallmarks of previously described fusion proteins (*El Omari et al., 2014*; *Flyak et al., 2018*; *Khan et al., 2015*; *Khan et al., 2014*; *Kong et al., 2013*; *Tzarum et al., 2019*), and may represent the prototype of a new class of fusion machinery. Therefore, understanding the conformational transitions and molecular triggers of E1E2 will reveal new biology and may guide HCV vaccinology; in the absence of a vaccine, HCV transmission continues at 1.5 million cases/year (as estimated by the *World Health Organization, 2022*).

HCV entry involves at least four host factors: CD81, scavenger receptor B-1 (SR-B1), claudin-1 and occludin. Additionally, epidermal growth factor receptor signalling contributes to receptor-complex formation and particle endocytosis, followed by endosomal, pH-dependent, fusion (*Baktash et al., 2018*; *Evans et al., 2007*; *Lupberger et al., 2011*; *Pileri et al., 1998*; *Ploss et al., 2009*; *Scarselli et al., 2002*). Current evidence indicates that only SR-B1 and CD81 interact directly with HCV, via the major glycoprotein E2, and that the minor glycoprotein, E1, contains the fusogen (*Hu et al., 2020*; *Ma et al., 2020*; *Perin et al., 2016*). Whilst there is a good structural understanding of the E2 ectodomain and partial characterisation of E1, how they assemble and function together is poorly understood (*Cao et al., 2019*; *Guest et al., 2021*). In particular, the molecular consequences of E1E2 interaction(s) with receptors and how this relates to the stepwise priming and triggering of the HCV fusion mechanism remains unknown.

Here, we demonstrate that genetic substitutions in E2 can switch HCV into a hyper-reactive state, this increases particle infectivity by enhancing the efficiency and kinetics of virus entry, but renders particles unstable and acutely sensitive to nAbs. This suggests a high propensity for inactivation of E1E2, presumably through a malfunctional refolding event or the premature triggering of fusion activity. Hyper-reactive HCV has a low dependency on SR-B1, indicating a role for this receptor in regulating the reactivity of E1E2. The N-terminal hypervariable region-1 (HVR-1) of E2 binds SR-B1 and we recently demonstrated that HVR-1 forms a disordered and dynamic protein tail (*Stejskal et al., 2020*). Molecular dynamic simulations and biophysical analysis reveal that hyper-reactive mutants exhibit stabilisation of HVR-1. This led us to hypothesise that HVR-1 exerts an autoinhibitory effect on E1E2 that is dependent on its dynamics, and stabilising HVR-1 (by receptor ligation or mutation) removes autoinhibition. Therefore, HVR-1 acts much like a safety catch on a firearm; preventing untimely triggering of E1E2 activity. Consistent with this, genetic deletion of HVR-1 switches HCV into a hyper-reactive state. Moreover, we provide evidence that antibody selection ensures that this safety catch mechanism remains engaged, maintaining low E1E2 reactivity and high resistance to nAbs.

## Results

### HCV explores evolutionary pathways to optimise virus entry

We established a continuous culture of HCV (J6/JFH HCVcc molecular clone) in Huh-7.5 cells and monitored viral evolution by next-generation sequencing (NGS). By day 42 we detected substitutions throughout the genome (before this, no mutations reached >5% frequency). The viral entry glycoproteins, E1E2, were particularly enriched for non-synonymous substitutions when compared to other coding regions (*Figure 1—figure supplement 1A*), suggesting adaptive optimisation of virus entry. These substitutions were largely located towards the C-terminus of E1 and the N-terminus of E2,

the latter being important for receptor and nAb interactions (*Figure 1—figure supplement 1B-D*; *Tzarum et al., 2018*). Some of the substitutions occurred at sites that are highly conserved in patient-derived HCV sequences (e.g. V371A, G406S, S449P), suggesting that in vitro replication does not recreate the evolutionary constraints of natural infection. We note that a critical difference between these settings is the complete absence of an adaptive immune response in vitro. From day 42 onward we observed the emergence of a mutant lineage with sequential fixation of I438V and A524T (found in the front layer and CD81 binding loop of E2, respectively, *Figure 1—figure supplement 1E*), and the concomitant loss of other variants from the population (e.g. I262L, *Figure 1—figure supplement 1E*). Introduction of these sequential mutations, by reverse genetics, resulted in a stepwise increase in HCV infectivity (*Figure 1A*). Next, we evaluated the entry pathway of these mutants by infecting cells in which the critical HCV entry factors (CD81, SR-B1, CLDN1, and OCLN) had been knocked out by CRISPR/Cas9 gene editing (*Figure 1B*). WT and mutant viruses were equally dependent on CD81, CLDN1, and OCLN, whereas the requirement for SR-B1 decreased in the mutant viruses in a manner that mirrored viral titre. This demonstrates that the efficiency of HCV entry is tightly linked to SR-B1 dependency.

So far, only SR-B1 and CD81 have been proven to interact directly with the HCV entry machinery, via E2 (*Pileri et al., 1998*; *Scarselli et al., 2002*). The precise molecular basis of E2-receptor interactions have yet to be defined at the structural level; nonetheless, mutational and antibody blocking experiments have demonstrated that SR-B1 binding occurs via the N-terminal HVR-1, whilst the CD81 binding site is thought to be composed of three discontinuous regions (antigenic site 412 [AS412], the front layer, and the CD81 binding loop) (*Kong et al., 2013*; *Kumar et al., 2021*; *Owsianka et al., 2006*; *Scarselli et al., 2002*). Previous work, including our own, suggests that SR-B1 is the initial receptor for HCV and is likely to prime subsequent stages of entry (including interaction with CD81) (*Augestad et al., 2020*; *Evans et al., 2007*; *Kalemera et al., 2019*). This is broadly analogous to the stepwise receptor-mediated priming of other viral fusion proteins (e.g. HIV, SARS-CoV-2); however, the molecular mechanism of SR-B1 priming of E1E2 was hitherto unknown.

We evaluated the relationship of WT and I438V A524T HCV with these receptors using a variety of techniques. First, we performed receptor blockade using antibodies targeting either SR-B1 or CD81 (*Figure 1C*; *Grove et al., 2017*; *Grove et al., 2007*). CD81 blockade prevented entry of both viruses equally - consistent with CD81 being an essential receptor - whereas the I438V A524T mutant was largely resistant to inhibition by anti-SR-B1. We corroborated this finding using BLT-4, a small molecule inhibitor of SR-B1 (*Figure 1—figure supplement 2*). Next, we used lentiviral vectors to over-express either receptor (*Figure 1D*). WT virus exhibited a strong response to increasing availability of SR-B1 and CD81, reaching a three- to fourfold enhancement relative to untreated parental cells. In contrast, I438V A524T HCV was less responsive to increased CD81 availability and was only marginally affected by SR-B1 over-expression. Finally, we directly evaluated E2-receptor interactions by measuring the binding of soluble E2 (sE2) to human SR-B1 or CD81 expressed on the surface of CHO cells (*Figure 1E & F*). I438V A524T sE2 exhibited altered receptor interactions, with >2 fold reduction in SR-B1 binding and >2 fold increase in CD81 interaction. In summary, HCV I438V A524T exhibits low dependency on SR-B1 and enhanced interactions with CD81. Notably, the reduction in SR-B1 binding by I438V A524T sE2 would suggest modulation of the SR-B1 binding site, found in HVR-1, even though neither of the mutated residues reside in this region.

We recently developed a mechanistic mathematical model of HCV entry, which can be used to explore entry by I438V A524T HCV (*Kalemera et al., 2019*). The model supports the notion that E1E2 can acquire CD81 via two routes: (1) SR-B1-mediated acquisition, where interaction with SR-B1 primes subsequent CD81 binding or (2) intrinsic acquisition of CD81, without the necessity for prior engagement of SR-B1 (*Figure 2—figure supplement 1A*). The latter pathway accounts for residual infection by WT virus in the absence of SR-B1 (*Figure 1B & C*). To faithfully represent the physical reality of virus entry, the model takes account of the intrinsic instability of HCV particles. We, therefore, compared the stability of WT and I438V A524T particles at 37 °C. Unexpectedly, we found that I438V A524T HCV was >3 fold less stable than WT virus (assessed by comparison of half-life, *Figure 2A*); this would suggest that the I438V A524T mutations increase the propensity for E1E2 to undergo spontaneous inactivation, such that particles become non-infectious more rapidly than WT; this is reminiscent of a hyper-reactive state observed in HIV-1 Env and a related phenotype in West Nile Virus (*Goo et al., 2017*; *Haim et al., 2011*).

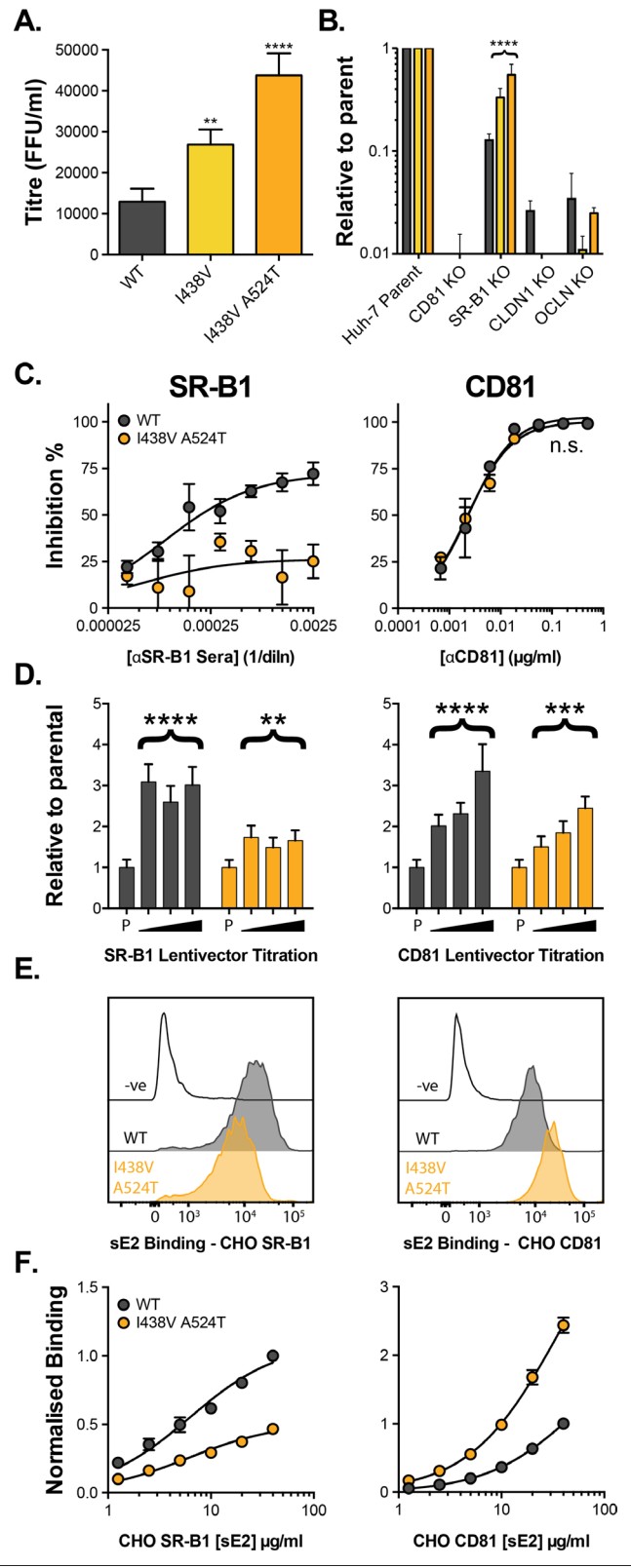

**Figure 1.** HCV evolves to optimise entry through altered receptor dependency. J6/JFH HCVcc mutants were isolated following continuous propagation in Huh-7.5 cells. (**A**) Infectivity of WT, I438V and I438V A524T HCVcc, expressed as foci forming units per ml. Values were normalised for input particle numbers. (**B**) WT and mutant HCV infection of parental Huh-7 cells and those CRISPR/Cas9 engineered to knock out the stated HCV entry

*Figure 1 continued on next page*

*Figure 1 continued*

factor. To aid direct comparison of each mutant, infection values have been expressed relative to that observed in parental Huh-7 cells. Values represent the mean of three independent experiments. Colour-coding as denoted in A. (**C**) Huh-7.5 cells were treated with anti-SR-B1 serum (left) or anti-CD81 mAb (right) to limit receptor availability. Infection by WT and I438V A524T HCVcc is expressed as % inhibition relative to infection of untreated cells. Data points represent the mean of three independent experiments. (**D**) To increase receptor availability, Huh-7.5 cells were transduced with a serial dilution of lentiviral vectors encoding either SR-B1 (left) or CD81 (right). Infection by WT and I438V A524T HCVcc is expressed relative to their respective infection of parental cells (**P**). Example data from one representative transduction is shown. (E. & F) CHO cells were transduced to express exogenous human SR-B1 (left) or CD81 (right), to which WT and mutant sE2 binding was assessed by flow cytometry. Upper plots (**E**). provide representative cytometry histograms of sE2 binding to transduced or parental (-ve) CHO cells. Lower plots (**F**). display quantification of sE2 binding, with values normalised to WT binding at 40 µg/ml. Data points represent the mean of three independent experiments. In all plots, error bars indicate standard error of the mean, asterisks denote statistical significance (ANOVA, GraphPad Prism). Curve fitting (C. and F.) performed with a hyperbola function and the curves compared to confirm significance difference (F-test, $p < 0.0001$, GraphPad Prism); n.s. denotes the lack of significant difference.

The online version of this article includes the following figure supplement(s) for figure 1:

**Figure supplement 1.** In vitro evolution of HCV.

**Figure supplement 2.** Inhibition of HCVcc infection by BLT-4.

We integrated the measurements of receptor dependency and particle stability (*Figures 1C–F and 2A*) into our mathematical model, allowing us to compare the entry characteristics of WT and I438V A524T HCV. We found that I438V A524T HCV adopts a hyper-reactive state to achieve efficient virus entry. Acquisition of CD81 via either route is enhanced (SR-B1-mediated acquisition, in particular, being ~1000 fold more efficient, *Figure 2—figure supplement 1B*, parameter $c_2$). Indeed, this would account for the ability of this mutant to tolerate reductions in SR-B1 availability (*Figure 1B & C*, *Figure 2—figure supplement 1*). Moreover, downstream entry events, encompassing cell surface translocation, endocytosis and fusion (*Baktash et al., 2018*) (which are not explicitly modelled here), were predicted to occur at ~sixfold higher rate (parameter e, *Figure 2—figure supplement 1B*). To explore this further we used the model to estimate the probability of entry by WT and mutant HCV upon varying availabilities of SR-B1 and CD81. Here I438V A524T HCV was predicted to achieve efficient entry over a broader range of receptor densities, for example at low CD81 availability (*Figure 2B*). An expected consequence of this is an increase in the kinetics of entry. We therefore used the model to estimate the relative speed at which WT and mutant HCV complete entry (*Figure 2C*) and, in parallel, made measurements of entry kinetics in vitro using synchronised HCV pseudoparticle (HCVpp) infection (*Haim et al., 2005*; *Figure 2D*). The model predicted that I438V A524T HCV enters >3 fold faster than WT. This estimate was in excellent agreement with the experimentally measured values, corroborating our modelling approach and confirming increased entry efficiency by I438V A524T HCV.

In summary, HCV is capable of evolving to optimise virus entry. It achieves this by adopting a hyper-reactive state, in which HCV exhibits low SR-B1 dependency, rapid acquisition of CD81 and increases in entry kinetics. However, the benefits of hyper-reactivity are somewhat offset by a decrease in stability; this likely reflects a greater propensity for E1E2 to undergo spontaneous and irreversible inactivation.

## Hyper-reactive HCV is acutely sensitive to all neutralising antibodies

The majority of HCV+ patients experience chronic life-long infection with persistently high viral loads; to achieve this, HCV must resist the E1E2-specific nAbs that arise in most individuals. Failure of HCV to evade and/or escape nAbs has been linked with viral clearance and understanding the molecular mechanisms of HCV antibody resistance is likely to inform ongoing HCV vaccinology (*Keck et al., 2018*; *Kinchen et al., 2019*; *Kinchen et al., 2018*).

We measured the sensitivity of HCV to chronic patient immunoglobulins (IgG). WT HCV was highly resistant to patient nAbs (*Figure 3A*); the neutralisation curve followed a log-linear relationship, such that successive 10-fold increases in IgG concentration yielded only a modest increase in neutralisation. In contrast, the hyper-reactive I438V A524T mutant was acutely sensitive to IgG, reaching complete neutralisation even at low concentrations.

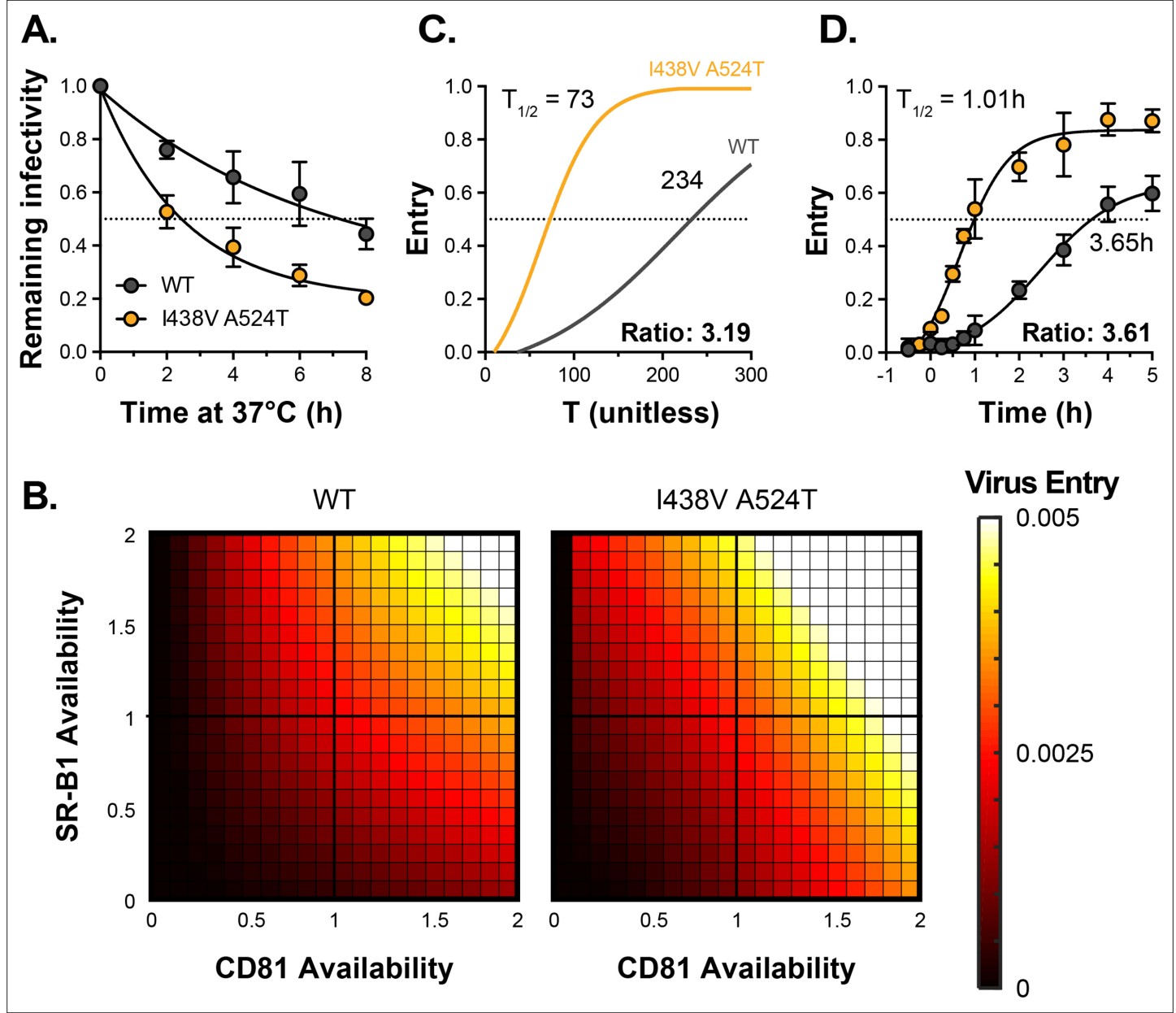

**Figure 2.** HCV I438V A524T is hyper-reactive. (**A**) WT and I438V A524T HCVcc were incubated for 0–8 hr at 37 °C before infection of Huh-7.5 cells. Remaining infectivity is expressed relative to t=0 time point. Data points represent the mean of two independent experiments, data was fitted using an exponential decay function. (**B**) Mathematical modelling was used to predict the entry characteristics of WT and I438V A524T HCVcc (also see *Figure 2—figure supplement 1*). Heat maps display the probability of virus entry (as denoted in the key), for any given virus particle, upon varying availability of SR-B1 or CD81. Receptor availability is scaled relative to parental Huh-7.5 cells (**C**) Kinetics of WT and mutant HCVcc entry, as predicted by mathematical modelling. The data is normalised to maximum entry. T represents uncalibrated time and, therefore, cannot be converted to real time, but relative differences can be estimated. Comparison of time to 50% entry suggests that I438V A524T HCVcc completes entry ~3 times faster than WT. (**D**) Kinetics of WT and mutant HCV entry were experimentally measured by synchronised infection of Huh-7.5 cells by HCVpp, followed by chase with a saturating inhibitory concentration of anti-CD81 mAb. Mutant HCVpp escaped the inhibitory effects of anti-CD81 ~3.5 times faster than WT. Data points represent the mean of three independent experiments, the data is normalised to entry in the absence of anti-CD81 mAb. The data was fitted with a sigmoid function. In all plots, error bars indicate standard error of the mean, fitted curves were confirmed to be significantly different (F-test, p<0.001, GraphPad Prism).

The online version of this article includes the following figure supplement(s) for figure 2:

**Figure supplement 1.** Mathematical modelling of HCV entry.

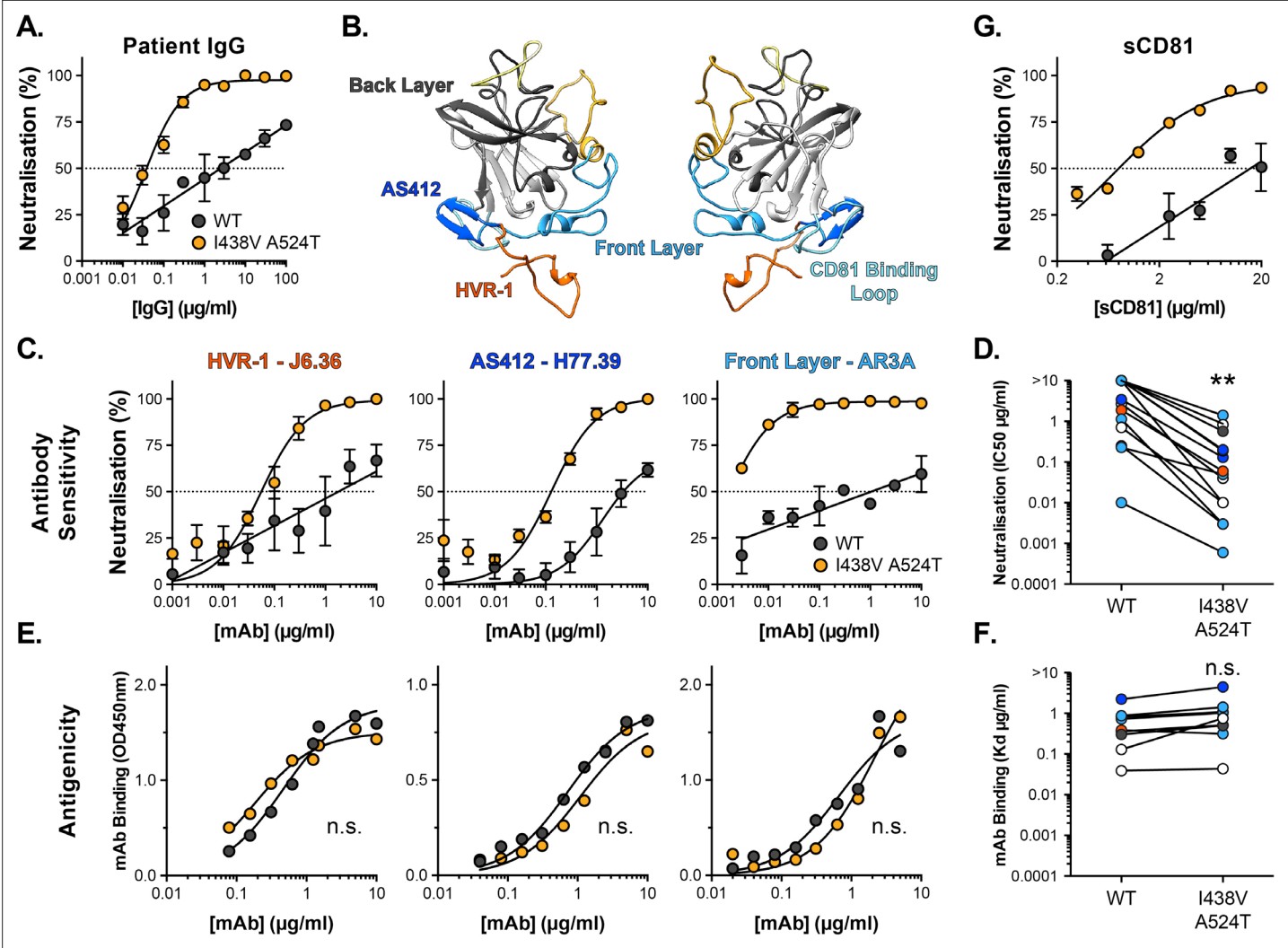

**Figure 3.** Hyper-reactive HCV is acutely sensitive to all neutralising antibodies. (**A**) Neutralisation of WT and I438V A524T HCVcc by a serial dilution of HCV patient IgG. (**B**) Molecular cartoon of major antigenic sites targeted by nAbs. (**C**) Neutralisation curves for three representative mAbs targeting distinct sites, the mAb name and specificity are provided and colour coded to match B. (**D**) 50% inhibitory concentrations of 14 mAbs, data points are colour coded according to their antigenic target, asterisks indicate statistical significance (T-test, GraphPad Prism). (**E**) Binding of mAbs to WT and I438V A524T E2 assessed by ELISA; mAbs matched to neutralisation data. (**F**) Estimated dissociation constants for 9 mabs, data points colour coded as above, there is no significant difference between WT and mutant (T-test, GraphPad Prism). (**G**) HCVcc neutralisation by soluble CD81 EC2. For all neutralisation curves, data points represent the mean of n=2 or 3 independent experiments. I438V A524T data were best fitted by a hyperbola function (GraphPad Prism), WT data were best fitted by a semi-log function (except for H77.39). For ELISA data, one representative experiment is shown, with data fitted with a hyperbola function. All neutralisation curves determined to be statistically significant (F-test, p<0.0001, Graphpad prism), there were no significant differences in the binding curves. In all plots, error bars indicate standard error of the mean. Estimates of IC50s and Kd were obtained from the fitted curves.

The online version of this article includes the following figure supplement(s) for figure 3:

**Figure supplement 1.** Antibody sensitivity and antigenicity of WT and I438V A524T HCV.

**Figure supplement 2.** Hyper-reactive phenotype is recapitulated in HCV pseudoparticles.

**Figure supplement 3.** pH sensitivity of HCV or Chikungunya (CHIKV) E1E2 pseudovirus .

Intense investigation of anti-HCV nAb responses, by others, has provided a detailed understanding of the major antigenic targets in E2 *Figure 3B*; for example, many potent nAbs target one or other component of the CD81 binding site (AS412, Front Layer and CD81 Binding Loop) (*Tzarum et al., 2018*). We measured the sensitivity of WT and mutant HCV to a panel of mAbs targeting a range of these antigenic targets (*Figure 3*, *Figure 3—figure supplement 1A*; *Bailey et al., 2017*; *Giang*

*et al., 2012*; *Pierce et al., 2016*; *Sabo et al., 2011*). Without exception, WT virus resisted neutral-isation whereas I438V A524T HCV was potently inhibited; this is best illustrated by examining the change in IC50 values across the panel, which demonstrates a~20-fold increase in sensitivity to mAbs (*Figure 3D*). We also assessed the ability of soluble CD81 EC2 to neutralise HCV (*Figure 3G*) and observed the same pattern of inhibition. To evaluate whether this global shift in nAb sensitivity reflected changes in antibody binding, we measured nAb interactions with sE2 by ELISA (*Figure 3E and F* and *Figure 3—figure supplement 1B*). Without exception, mAbs bound equally to WT and I438V A524T E2, suggesting that WT and mutant E2 are antigenically equivalent.

A unique feature of HCV particles is their association with host-derived lipoprotein components, resulting in so-called lipo-viro-particles. Lipoprotein receptors and cholesterol transporters have been implicated in HCV entry and antibody evasion (*Ding et al., 2014*; *Grove et al., 2008*; *Lindenbach and Rice, 2013*); indeed, SR-B1 is a high-density lipoprotein receptor and it has been suggested to interact with host apolipoproteins on HCV virions (*Maillard et al., 2006*). Nonetheless, I438V A524T hyper-reactivity (low SR-B1 dependency, instability and nAb sensitivity) was preserved on HCVpp (*Figure 3—figure supplement 2*), which do not exhibit lipoprotein associations (*Hsu et al., 2003*). Therefore, the hyper-reactive phenotype is intrinsic to E1E2 and does not require contribution from host lipoproteins components.

HCV entry is sensitive to inhibitors of endosomal acidification and, therefore, the conformational changes necessary for HCV fusion are thought to require a low-pH trigger (*Bartosch et al., 2003*; *Blanchard et al., 2006*; *Hsu et al., 2003*; *Meertens et al., 2006*). However, unlike many classically acid-dependent viruses, HCV is resistant to deactivation by low pH treatment (*Meertens et al., 2006*; *Tscherne et al., 2006*). This suggests additional priming events are required to unlock the pH-dependent steps in HCV fusion; indeed binding of CD81 has previously been shown to sensitise HCV to acidic buffers (*Sharma et al., 2011*). Given the stability and neutralisation phenotypes of the hyper-reactive HCV mutant, we reasoned that it may exhibit altered sensitivity to low-pH treatment. To test this we immobilised HCVpp for incubation at pH 7, 6, and 5, prior to infection of Huh-7 cells (*Sharma et al., 2011*). Both WT and I438V A524T pseudoparticles were resistant to acid treatment, whereas control pseudoparticles bearing Chikungunya E1E2 were potently inactivated (*Figure 3—figure supplement 3A, B*). Next, to examine receptor-mediated priming of HCV pH sensitivity, we repeated the experiment with the addition of soluble CD81 ectodomain (sCD81). Here, WT pseudoparticles remained acid-resistant, whereas I438V A524T displayed a CD81-dependent sensitivity to treatment at both pH 6 and 5 (*Figure 3—figure supplement 3C*). These data are consistent with the notion of CD81 as a gate-keeper for pH-dependent changes in E1E2 and indicate that I438V A524T HCV is more sensitive to this priming mechanism.

In summary, I438V A524T HCV exhibits acute sensitivity to antibody-mediated neutralisation without intrinsic changes to the antigenicity of E2. This would suggest that whilst antibody binding remains unaltered, the ability of HCV to resist neutralisation has fundamentally changed. This likely reflects the hyper-reactive state of I438V A524T E1E2; much like incubation at 37 °C (*Figure 2A*), nAb interactions may trigger irreversible inactivation of E1E2. Alternatively, the accumulation of inactive glycoproteins in hyper-reactive HCV (evidenced by thermal instability; *Figure 2A*), may render HCV particles incapable of tolerating further E1E2 inactivation by nAbs. Moreover, I438V A524T E1E2 was sensitive to CD81-dependent low-pH inactivation, further suggesting that I438V A524T HCV is more prone to irreversible inactivation by ligand-mediated and biochemical cues.

## Hyper-reactive HCV exhibits stabilisation of HVR-1

Antigenic similarity indicates there is no gross conformational change in I438V A524T E2. However, when subjected to limited proteolysis I438V A524T E2 undergoes cleavage more rapidly than WT E2 (*Figure 4—figure supplement 1*), this suggests that subtle structural differences may underpin hyper-reactivity. We have previously used molecular dynamic simulations (MD) to explore the conformational landscape of E2, finding that flexibility and disorder are conserved features of E2, consistent with other reports (*Balasco et al., 2018*; *Kong et al., 2016*; *Meola et al., 2015*; *Stejskal et al., 2020*; *Ströh et al., 2018*; *Vasiliauskaite et al., 2017*). In particular, we discovered that the N-terminus of E2, HVR-1 (containing the SR-B1 binding site), is a putative intrinsically disordered peptide tail. We used MD to examine the motion of E2, performing five independent 1μs simulations of WT and I438V A524T. The overall dynamics of either E2 were similar, as reflected in their root mean square

fluctuation profiles (RMSF; *Figure 4C*). However, the HVR-1 tail of I438V A524T E2 exhibited consistent stabilisation (*Figure 4*), this is best illustrated by root mean square deviation (RMSD), which captures motion over time (*Figure 4A and B*, *Figure 4—figure supplements 2 and 3*), and RMSF of HVR-1 (*Figure 4C and D*).

Given these data, we reasoned that I438V A524T E2 may share biophysical characteristics with E2 lacking HVR-1 (ΔHVR-1, in which HVR-1 is genetically deleted). We first analysed sE2 by circular dichroism spectroscopy (CD), which provides a low-resolution ensemble measurement of protein structure (*Figure 4—figure supplement 4*). Here, I438V A524T and ΔHVR-1 E2 were biophysically indistinguishable from one another but distinct from WT E2. In particular, estimation of the structural composition of I438V A524T and ΔHVR-1 E2 indicated a significant reduction of unordered components *Figure 4E*; this is broadly consistent with our MD experiments. We also assessed sE2 by nano differential scanning fluorimetry (nanoDSF), which exploits the changes in intrinsic protein fluorescence upon solvent exposure of tryptophan residues to provide a surrogate measure of protein folding and to determine melting temperature (*Figure 4—figure supplement 5*). Consistent with other reports, the apparent melting temperature of sE2 was high (>80 °C; *Figure 4—figure supplement 5B*; *Kong et al., 2016*), this likely represents unfolding of the globular core of E2 and, in this respect, we observed no differences between each E2. However, at physiological temperatures the intrinsic fluorescence ratio of I438V A524T and ΔHVR-1 E2 were higher than WT (*Figure 4 E*, *Figure 4—figure supplement 5A*), consistent with I438V A524T and ΔHVR-1 E2 being biophysically distinct from WT. The location of tryptophan residues is relevant when interpreting nanoDSF data; in this regard, HVR-1 is directly upstream of a highly conserved tryptophan residue in AS412 that has been previously demonstrated to be important for CD81 binding (W420 *Cowton et al., 2016*; *Owsianka et al., 2006*).

In summary, MD simulations suggest that E1E2 hyper-reactivity is associated with reduced HVR-1 dynamics. This is supported by the biophysical similarity of mutant and ΔHVR-1 E2. Therefore, HVR-1 dynamics provide a potential mechanism linking E1E2 reactivity, SR-B1 receptor dependency, entry efficiency and nAb sensitivity.

## Hypervariable region-1 is an entropic safety catch

Based on our data, we hypothesised that HVR-1 exerts an autoinhibitory effect on E1E2 that is dependent on its dynamics and conformational entropy. Moreover, during virus entry, engagement of SR-B1 will, necessarily, constrain HVR-1, reduce conformational entropy and turn off autoinhibition. Therefore, HVR-1 is analogous to a safety catch on a firearm that suppresses the reactivity of E1E2 to prevent untimely or inappropriate triggering of the HCV entry mechanism. SR-B1 turns off the safety catch at the cell surface to prime virus entry. In the hyper-reactive I438V A524T mutant, pre-stabilisation of HVR-1 turns off the safety catch prematurely, reducing the requirement for SR-B1 and enhancing entry efficiency, but rendering E1E2 more prone to spontaneous inactivation: I438V A524T HCV is on a hair-trigger. The entropic safety catch model is summarised in *Figure 5A*.

Tuning of protein function by disordered peptide tails has been described in other systems (*Uversky, 2013*); in these cases, the tail can generate an entropic force which acts upon the rest of the protein (*Keul et al., 2018*). We examined this possibility in E1E2 using dynamic cross-correlation (DCC), which provides a measure of correlated/coordinated motion within MD simulations. DCC analysis of WT E2 suggests that motions within HVR-1 are transmitted throughout the protein (*Figure 5B*), consistent with an exerted force. In the simulations of I438V A524T E2, HVR-1 is stabilised (*Figure 4*) and these correlated motions are absent (*Figure 5B*), suggesting the lack of an entropic force. This provides a potential molecular mechanism by which HVR-1 dynamics can influence the entire protein.

Given these observations, we reasoned that genetic deletion of HVR-1 would remove this entropic force, turn off the safety catch and confer a hyper-reactive phenotype. We characterised the reactivity of WT, I438V A524T, and ΔHVR-1 HCV by measuring infectivity, receptor dependency, stability and nAb sensitivity (*Figure 5C–F*). In each case, ΔHVR-1 HCV exhibited a hyper-reactive phenotype above and beyond that of I438V A524T HCV; ΔHVR-1 HCV has very high infectious titres, low SR-B1 dependency, high instability and acute sensitivity to patient IgG. These data are completely consistent with the entropic safety catch model.

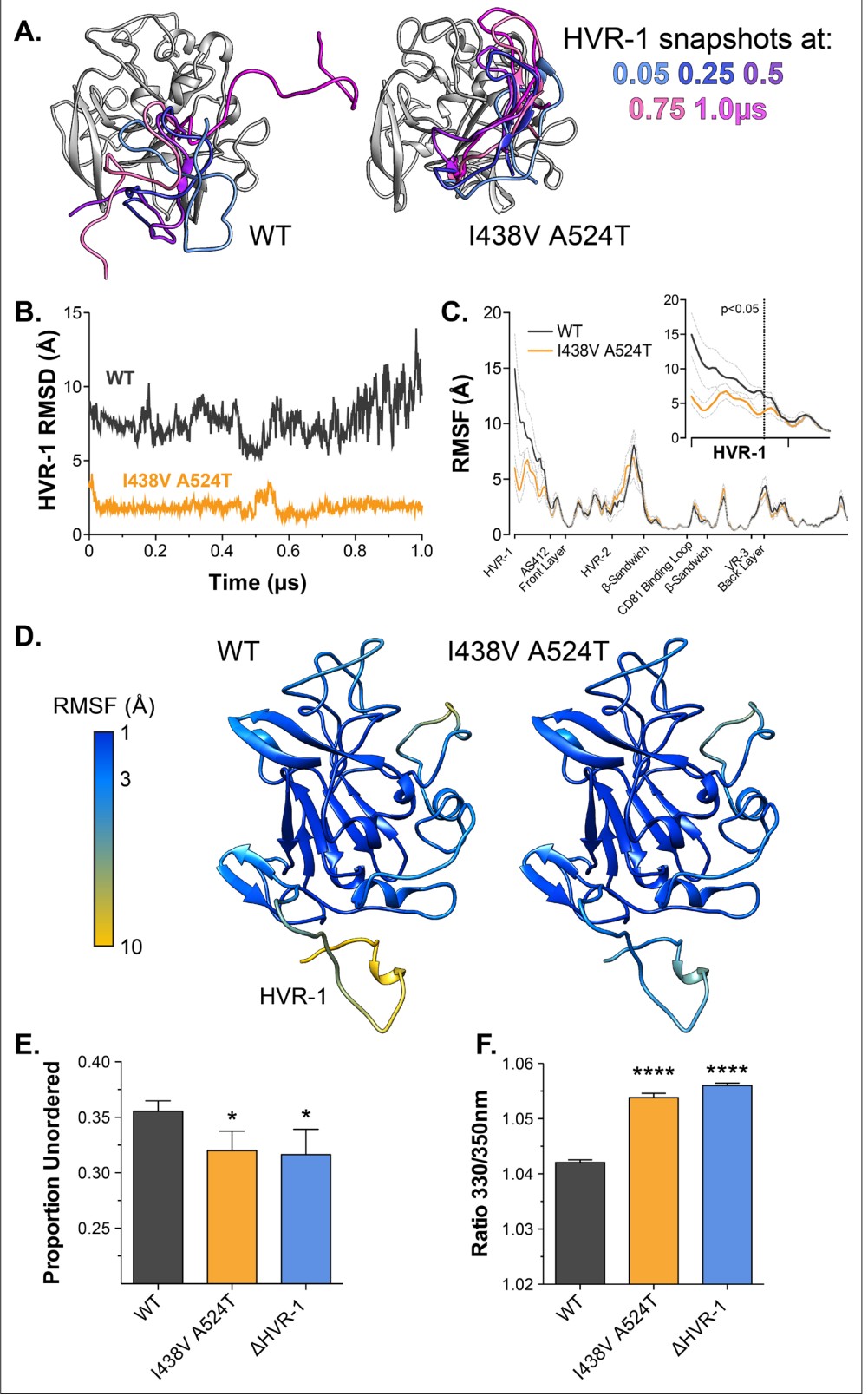

**Figure 4.** Hyper-reactive HCV exhibits stabilisation of HVR-1. The conformational dynamics of WT and I438V A524T E2 ectodomain were explored by five independent 1μs MD simulations. (**A**) Images summarising two representative simulations; superimposed snapshots of HVR-1 are colour coded according to time (as shown in key), the remainder of E2 is shown in grey for the earliest frame only. (**B**) Root mean square deviation (RMSD) of

*Figure 4 continued on next page*

*Figure 4 continued*

HVR-1 for the simulations shown in A. (**C**) Average root mean square fluctuation (RMSF) for WT and I438V A524T E2, from five independent experiments, error bars indicate standard error of the mean. X-axis denotes regions of E2. Inset provides a zoom of the data for HVR-1, RMSF values to the left of the dashed line reach statistical significance (ANOVA, GraphPad Prism). (**D**) Average RMSF values (as in C.) plotted on to molecular cartoons of E2, colour coded as shown in key. HVR-1 is labelled on the WT structure. Summaries of MD simulations are provided in *Figure 4—figure supplements 2 and 3*. (**E**) Estimation of unordered protein content, by circular dichroism spectroscopy, for WT, I438V A524T and ΔHVR-1 sE2. Data represent the mean of three independent measurements. (**F**) Intrinsic fluorescence ratio (330 nm over 350 nm), measured by nano differential scanning fluorimetry, for WT, I438V A524T, and ΔHVR-1 sE2 at 37 °C. Data represent the mean of three independent measurements. Asterisks denote statistical significance (ANOVA, GraphPad Prism).

The online version of this article includes the following source data and figure supplement(s) for figure 4:

**Figure supplement 1.** Limited proteolysis of soluble E2.

**Figure supplement 1—source data 1.** Raw unedited blot images (A & C) and annotated blots indicating band and sample identities (B & D) for WT (A & B) and I438V A524T E2 (C & D).

**Figure supplement 2.** Five independent 1μs MD simulations of WT E2.

**Figure supplement 3.** Five independent 1μs MD simulations of I438V A524T E2.

**Figure supplement 4.** Biophysical analysis of E2 by circular dichroism spectroscopy.

**Figure supplement 5.** Biophysical analysis of E2 by nano differential scanning fluorimetry.

## Consistent emergence of hyper-reactive mutants in the absence of antibody selection

Given the high infectivity of I438V A524T and ΔHVR-1 HCV, there remains a question over the advantages of evolving a mechanism to suppress E1E2 activity. The likely explanation is that in chronic infection nAbs necessitate tight control over E1E2 reactivity and that antibody selection prevents the emergence of hyper-reactive mutants. In contrast, cell culture replication is devoid of adaptive immunity and may permit the emergence of viruses with optimised entry but compromised nAb evasion. To examine this, we first returned to our original cell culture adaptation experiment. At day 42 multiple minor variants co-existed with the subsequently successful lineage *Figure 1—figure supplement 1*; one of these mutations S449P (10.2% frequency) has previously been suggested to alter SR-B1 dependency (*Zuiani et al., 2016*). We, therefore, reasoned that S449P HCV may exhibit a hyper-reactive phenotype. In every respect, S449P HCV was indistinguishable from I438V A524T HCV (*Figure 6A–D*). Moreover, MD simulations of S449P E2 demonstrated stabilisation of HVR-1 to the same extent as I438V A524T (*Figure 6E–F*, *Figure 6—figure supplement 1*). This suggests that, in the absence of immune selection, HCV explores independent evolutionary pathways towards the entry-optimised hyper-reactive phenotype.

To test the importance of nAb selection, we performed new culture adaptation experiments in the presence or absence of patient IgG and measured the infectivity and antibody sensitivity of the resultant virus populations (*Figure 6G and H*). The antibody selected culture exhibited low infectivity and high resistance to neutralisation, much like WT HCV. In contrast, the culture without selection developed hallmarks of the hyper-reactive phenotype; having increased infectivity and high sensitivity to neutralisation. NGS analysis of the culture with IgG selection revealed a mixed population of WT (>75%) and minor variants (<25% in total), suggesting limited adaptation, whereas the culture without selection was dominated (>80%) by a triple E1E2 mutant: R317H, T387A, R408G (*Figure 6I*). Notably, the latter two of these mutations are found within HVR-1. We characterised R317H, T387A, R408G E1E2 using the HCVpp system finding it to have nAb sensitivity and SR-B1 independence above that of I438V A524T E1E2 (*Figure 6J & K*).

These data strongly support the HVR-1 entropic safety catch model and suggest that emergence of hyper-reactive HCV can only occur in the absence of antibody selection. Therefore, the HVR-1 safety catch likely represents an important mechanism for maintaining HCV resistance to neutralising antibodies. If this is the case, we would expect the safety catch model to hold true in other HCV clones (thus far our experiments have been limited to genotype 2a J6 E1E2). To test this, we introduced analogous L438V A524T mutations into genotype 1a H77 E1E2 for characterisation using HCVpp. In line with our expectations, this resulted in a >10-fold increase in sensitivity to patient IgG and a

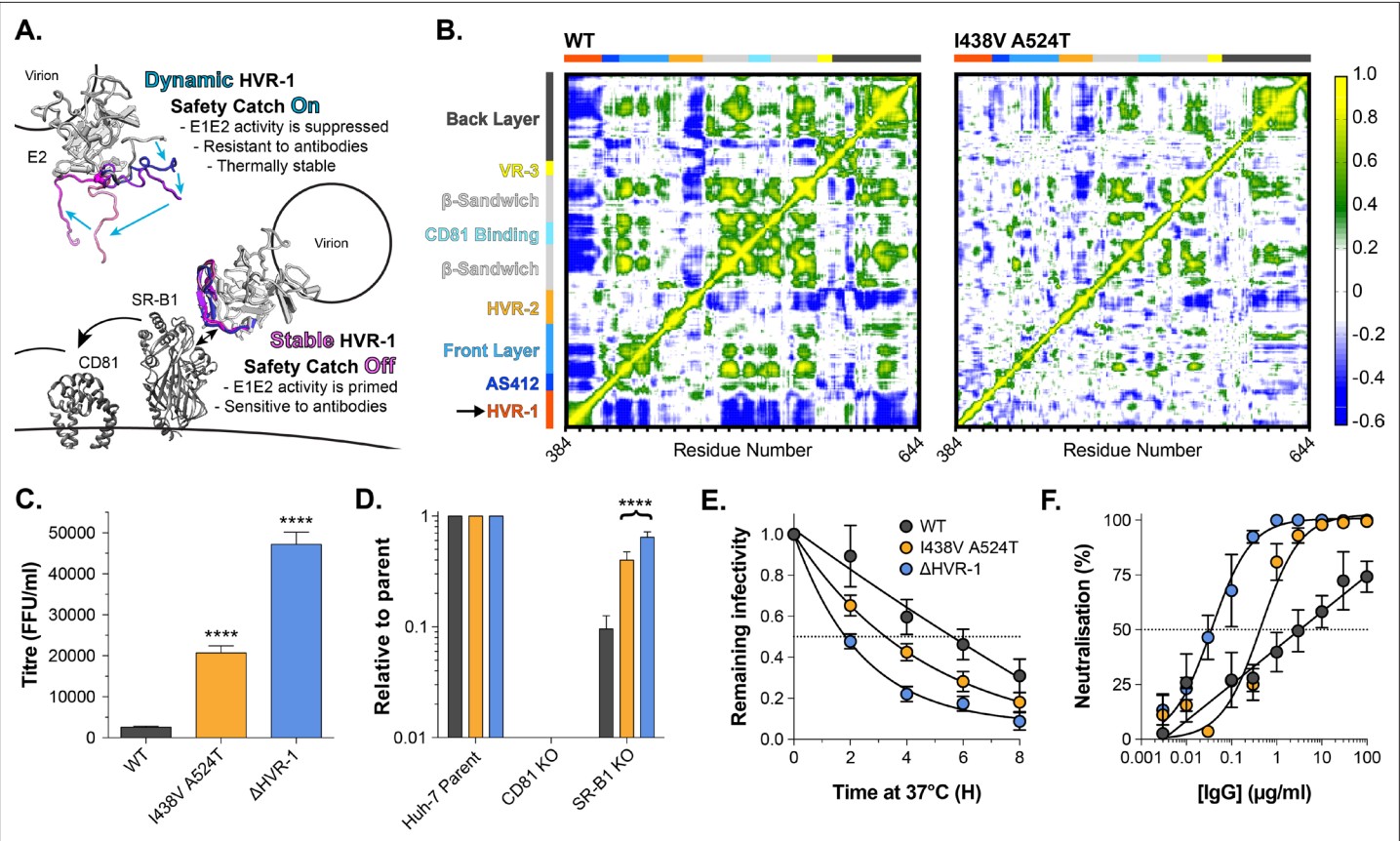

**Figure 5.** HVR-1 is an entropic safety catch. (**A**) The entropic safety catch model. On free virions HVR-1 is dynamic and it acts as an autoinhibitory safety catch on E1E2 function. Interactions with SR-B1 at the cell surface stabilise HVR-1, turning off the safety catch and enabling virus entry. (**B**) Dynamic cross correlation provides a residue-by-residue pairwise comparison of motion in MD trajectories to reveal correlations/anti-correlations in protein movement. Average DCC matrices for WT and I438V A524T E2 MD simulations are provided. As depicted in the key, yellow/green indicate positive correlations; blue indicates negative correlations; white indicates lack of correlation. (**C**) Mean infectivities of WT, I438V A524T and ΔHVR-1 HCVcc; foci forming units are corrected for input particle numbers (**D**) HCVcc infection of parental Huh-7 cells or those CRISPR/Cas9 edited to prevent expression of CD81 or SR-B1. Data is expressed relative to parental cells, mean of three independent experiments. (**E**) Stability of HCVcc at 37 °C, data points represent the mean of three independent experiments, values normalised to infection at t=0, data was fitted using an exponential decay function. (**F**) Neutralisation of HCVcc by patient IgG, data points represent the mean of three independent experiments. Data was fitted with a hyperbola function (I438V A524T and ΔHVR-1) or semilog function (WT). In each plot, error bars indicate standard error of the mean; asterisks denote statistical significance (ANOVA, Graphpad prism); all curves determined to be statistically significant (F-test, p<0.0001, Graphpad prism).

~fourfold decrease in SR-B1 dependency (*Figure 6L & M*). Moreover, MD simulations indicate that H77 L438V A524T E2 exhibits decreased HVR-1 dynamics when compared to WT H77 (*Figure 6N*). These data are consistent with the HVR-1 safety-catch mechanism being conserved across diverse HCV genotypes.

## Discussion

The entropic safety catch model, presented here, reconciles numerous reports linking HCV receptor dependency, HVR-1, particle infectivity and nAb sensitivity (*Augestad et al., 2020*; *Bankwitz et al., 2014*; *Bankwitz et al., 2010*; *Bitzegeio et al., 2010*; *Dhillon et al., 2010*; *Grove et al., 2008*; *Kalemera et al., 2019*; *Keck et al., 2009*; *Koutsoudakis et al., 2012*; *Lavie et al., 2014*; *Prentoe et al., 2019*; *Prentoe et al., 2016*; *Prentoe et al., 2014*; *Prentoe et al., 2011*; *Song et al., 2012*) and, therefore, provides mechanistic clarity on the early events of HCV entry. We demonstrate that auto-inhibition by the entropic safety catch can be turned off through substitutions that stabilise HVR-1, permitting increases in virus entry efficiency to be traded for decreases in nAb resistance. This switch-ability will likely synergise with the genetic plasticity of HCV, allowing rapid response to changing

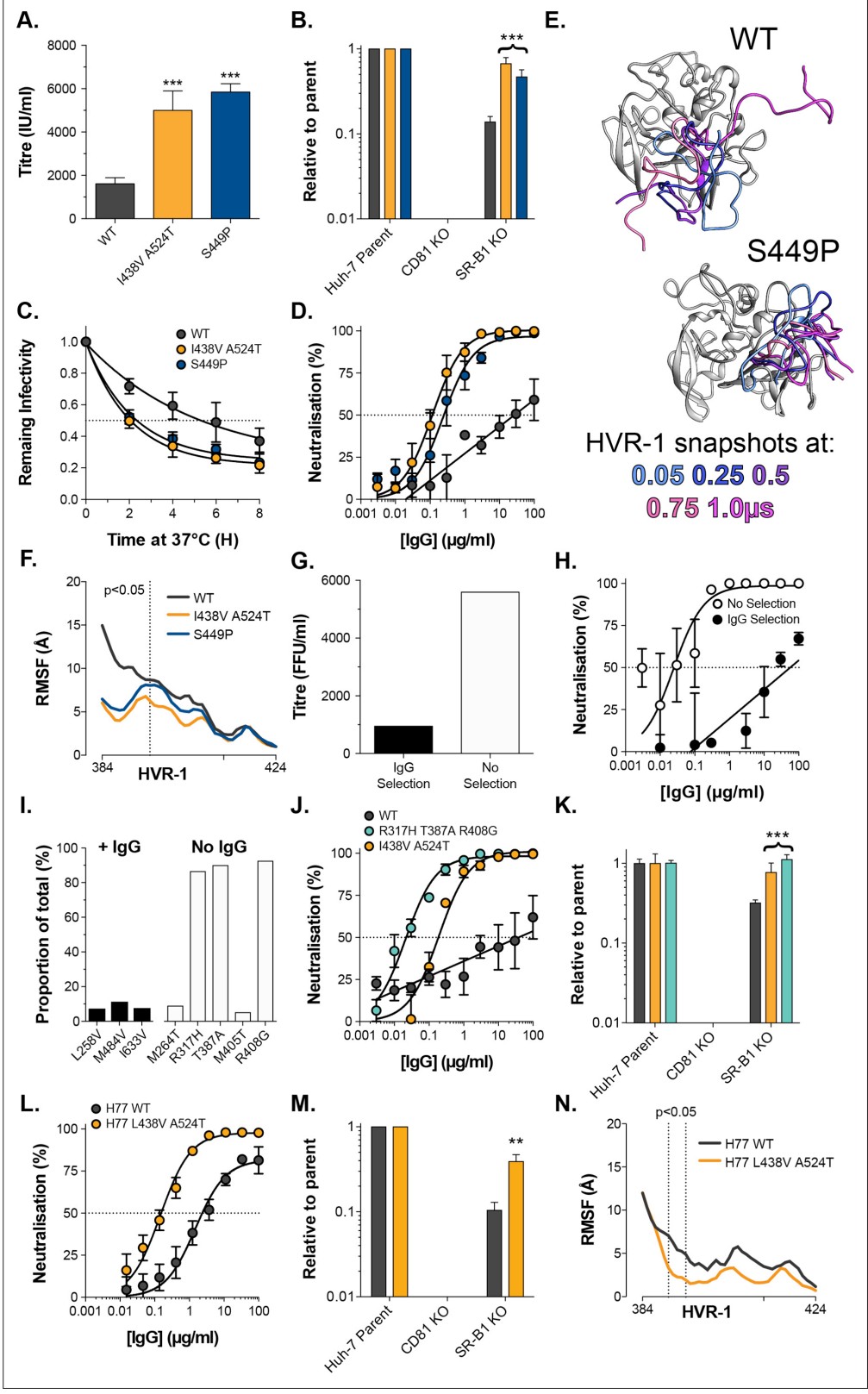

**Figure 6.** Consistent emergence of hyper-reactive mutants in the absence of antibody selection. (**A**) Mean infectivities of WT, I438V A524T and S449P HCVcc; foci forming units are corrected for input particle numbers (**B**) HCVcc infection of parental Huh-7 cells or those CRISPR/Cas9 edited to prevent expression of CD81 or SR-B1. Data is expressed relative to parental cells, mean of three independent experiments, colour coding as

*Figure 6 continued on next page*

*Figure 6 continued*

in panel 6A. (**C**) Stability of HCVcc at 37 °C, data points represent the mean of three independent experiments, values normalised to infection at t=0, data was fitted using an exponential decay function. (**D**) Neutralisation of HCVcc by patient IgG, data points represent the mean of three independent experiments. Data was fitted with a hyperbola function (I438V A524T and S449P) or semilog function (WT), colour coding as in panel 6D. (**E**) Images summarising representative MD simulations of WT and S449P E2 ectodomain; superimposed snapshots of HVR-1 are colour coded by time, as denoted in the key, the remainder of E2 is shown in grey for the earliest time point only. (**F**) Average RMSF of WT, I438V A524T and S449P HVR-1; values to the left of the dashed line reach statistical significance for both mutants compared to WT (ANOVA, GraphPad Prism). (**G**) & (**H**) J6/JFH HCVcc was propagated in Huh-7.5 cells with and without antibody selection by HCV + patient IgG (see Methods). (**G**) Infectious titre of HCVcc after propagation with and without antibody selection. (**H**) HCV + patient IgG neutralisation of HCVcc cultured with and without antibody selection, data points represent the mean of three technical repeat, data was fitted with a hyperbola function (No selection) or semilog function (IgG selection). (**I**) NGS analysis of viral cultures; plots display identity and frequency of variants (>5%) in IgG selected (black bars) and unselected cultures (white bars). (**J**) Neutralisation sensitivity of HCVpp bearing WT, I438V A524T, or R317H T387A R408G E1E2, assessed as in panel 6D. (**K**) Receptor dependency of HCVpp, assessed as in panel 6B, colour coding as in panel 6J. (**L**) Neutralisation sensitivity of HCVpp bearing H77 clone WT or L438V A524T E1E2, assessed as in panel 6D (**M**) Receptor dependency of HCVpp, assessed as in panel 6B, colour coding as in panel 6L. (**N**) Average RMSF of H77 WT or L438V A524T E2 HVR-1 from five independent MD simulation; values within dashed lines reach statistical significance (ANOVA, GraphPad Prism). In each plot, error bars indicate standard error of the mean; asterisks denote statistical significance (ANOVA, Graphpad prism); all curves determined to be statistically significant (F-test, p<0.0001, Graphpad prism).

The online version of this article includes the following figure supplement(s) for figure 6:

**Figure supplement 1.** Five independent 1µs MD simulations of S449P E2.

---

selection pressures, and may offer explanation for HCV's ability to maintain infection in the face of continual immune assault. Indeed, various studies have identified patient-derived E2 polymorphisms (including position 438) that alter antibody sensitivity, receptor usage and infectivity (*Bailey et al., 2015*; *El-Diwany et al., 2017*; *Fofana et al., 2012*; *Keck et al., 2009*); these observations are consistent with changes in E1E2 reactivity mediated via the safety catch.

HVR-1 is, itself, an immunodominant target for antibodies; however, anti-HVR-1 responses are rapidly evaded through mutational escape (*von Hahn et al., 2007*). Our study would suggest that disorder and conformational entropy are critical to HVR-1 function; this feature likely underpins its ability to tolerate extensive variation. Moreover, whilst antigenic variation is detrimental to other functionally important regions of E2 (e.g. the CD81 binding site *Keck et al., 2009*; *Kinchen et al., 2018*), continuous antigenic drift of HVR-1 likely ensures that disorder is maintained. This provides a potential feedback system where nAb selection promotes conformational entropy, keeping the safety catch mechanism engaged and, effectively, creating a molecular sensor for antibody selection.

Our data suggest linkage (either direct or through allostery) between HVR-1 and distant regions of E2 (*Figure 5B*); however, we do not yet have a complete molecular understanding of the mechanism of E1E2 autoinhibition by HVR-1. Moreover, we were unable to identify consistent interactions between HVR-1 and other regions of E2 that could explain stabilisation in the mutants. Future investigations will aim to understand how mutations, and E2-SR-B1 interactions, toggle the HVR-1 safety catch and identify the structural/dynamical consequences for the E1E2 complex. Relevant to this, *Augestad et al., 2020* have recently suggested that SR-B1 may trigger a conformational opening of E1E2, associated with changes in AS412 (a functional epitope directly downstream of HVR-1); although the structure of this 'open' state, and how it relates to E1E2 reactivity, remains unclear. Alternatively, *Tzarum et al., 2020* recently identified a conformational switch in the front layer of E2 that may be important for CD81/antibody engagement; might this be regulated by the safety catch? Ultimately, turning off the entropic safety catch is likely to be the first step towards activating the fusion activity of E1E2 (although our data suggest further molecular cues are necessary to prime pH sensing); we expect this insight to guide future advances in elucidating the fusion mechanism of HCV.

Beyond HCV, we propose that this work has broader implications. The extent to which structural disorder regulates protein function is only just becoming apparent (*Ferreon et al., 2013*; *Keul et al., 2018*; *Wright and Dyson, 2015*). We expect that evolution has bestowed similar, as yet undiscovered, entropic switches on other viruses. Indeed, the N-terminal preS1 domain of the HBV entry

protein is disordered and harbours a receptor binding site (*Chi et al., 2007*), the C-terminal tail of HIV gp41 contains disordered components and regulates Env reactivity (*Steckbeck et al., 2013*; *Uversky, 2013*) and disordered loops in the N-terminal domain of SARS-CoV-2 spike modulate protein function (*Dicken et al., 2021*; *Meng et al., 2021*). Tunable entropy may yet prove to be a common strategy for navigating the thermodynamic landscapes of virus entry and antibody evasion.

## Materials and methods

### Cell cultures

Huh-7.5 cells were acquired from Apath LLC. CRISPR Cas9 receptor KO Huh-7 cells (*Yamamoto et al., 2016*), and parental Huh-7 cells, were generously provided by Yoshiharu Matsuura (Osaka University, Japan). CHO and HEK293T were acquired from the American Type Culture Collection. Cell line identities were confirmed by STR profiling, and all cell lines tested negative for the presence of mycoplasma. All cells were grown at 37 °C in Dulbecco's Modified Eagle Medium (DMEM) supplemented with 10% foetal calf serum (FCS), penicillin and streptomycin and non-essential amino acids.

### Antibodies

Mouse Anti-NS5 mAb (S38) and anti-CD81 mAb (2.131) were a gift from Prof. Jane McKeating (University of Oxford). Rabbit Anti-SR-B1 serum was provided by Dr. Thierry Huby (INSERM, Paris). Mouse anti-E2 mAbs J6.36 and H77.39 were a gift from Michael Diamond (Washington University). Human anti-E2 mAbs were kindly provided by Dr. Mansun Law (SCRIPPS, La Jolla) (AR2A, AR3A, AR3C, AR4A, AR5A), Dr. Steven Foung (Stanford) (HC33.1.53, HC84.26, CBH-7, HC1, CBH-4B, CBH-7, CBH-23), and Prof. James Crowe (Vanderbilt University) (HepC3, HepC43). StrepMAB-classic was purchased from IBA Lifesciences (Göttingen, Germany).

### Isolation of patient IgG

Blood samples were collected from HCV + patients under ethical approval: "Characterising and modifying immune responses in chronic viral hepatitis"; IRAS Number 43993; REC number 11/LO/0421. Extracted serum was heat inactivated, filtered and diluted 1:1 in PBS. Total IgG was captured using a HiTrap protein G column (Cytiva, MA, USA), eluted in pH2.7 glycine buffer and buffer exchanged into PBS. For experiments, batches of pooled IgG were created by equimolar combination of IgG from two patient samples.

### Production of HCVcc

Plasmid encoding cell-culture proficient full-length J6/JFH-1 (acquired from Apath LLC) was used as a template for the in vitro production of infectious HCV RNA (*Lindenbach et al., 2005*). To initiate infection, viral RNA was introduced into Huh-7.5 cells using a BTX830 electroporator (Harvard Instruments, Cambridge, UK). From 3 to 7 days post electroporation, cell culture supernatants containing infectious J6/JFH-1 HCVcc were harvested every 3–5 hr. Short harvest times limit the opportunity for virus decay thereby preventing the accumulation of non-infectious particles. To ensure maximum reproducibility between experiments, a standardised stock of experimental virus was generated by pooling the harvested supernatants.

### HCVcc adaptation experiments

In the initial adaptation experiment (*Figure 1*), a continuous culture of J6/JFH HCVcc was established in Huh-7.5 cells. Infected cells were passaged, twice weekly, with an excess of uninfected target cells added whenever the culture reached 90–100% infection. This proceeded for 20 weeks. For the adaptation experiments with and without IgG selection (*Figure 6*) we adopted a serial passage strategy. Huh-7.5 cells were electroporated with J6/JFH genomic RNA and the next day medium was replaced with DMEM 3% FCS plus or minus HCV[+] patient IgG at 100 µg/ml. Once 90–100% of cells were determined to be infected, the culture supernatant was harvested, clarified by centrifugation, and used to infect fresh Huh-7.5 cells. Typically, this was done every week. We performed six rounds of passage selection. In either case (continuous culture or passage) cell culture supernatants were frozen at –20 °C at regular intervals for analysis by NGS.

## HCVcc infections

Huh-7.5 and receptor KO Huh-7 cells were seeded at $1.5 \times 10^4$ cells per well of a 96 well plate 24 hr prior to the experiment. To quantify infectious titres, cells were challenged with a twofold serial dilution of virus stock (1/4 to 1/64) in DMEM 3% FCS (infection medium). Cells were incubated with viral supernatants for five hours before the cells were subsequently washed, and fresh medium was added. Infections were allowed to proceed for 48 (Huh-7.5 cells) or 72 (Huh-7 lines) hr before reading out. HCVcc replication was quantified by fluorescence microscopy. Cells were fixed with 100% methanol and stained for viral NS5A protein with mAb S38. HCV foci-forming units (FFU) were quantified by manual counting, or percentage infection determined by automated analysis of fluorescence micrographs (*Culley et al., 2016*).

## Neutralisation/inhibition assays

Huh-7.5 cells were seeded, as above. For neutralisation assays, virus was preincubated with a dilution series of mAb or soluble CD81 EC2 for 1 hour at 37 °C prior to infection. For anti-receptor blockade experiments, Huh-7.5 cells, seeded for infection, were pre-incubated at 37 °C with 50 µl DMEM 3% FCS containing a serial dilution of either rabbit-SR-B1 serum or mouse anti-CD81 mAb 2.131. One hour later, wells were challenged with virus. In each case the infections were processed as described above.

## Next generation sequencing and analysis

RNA was extracted from cell culture supernatants containing HCVcc by a BioRobot MDx instrument using QIAamp Virus BioRobot MDx Kits. Extracted RNA samples were amplified as described (*Aisyah et al., 2019*) processed locally within the UCL Hospital Virology laboratories for PCR library preparation and Next Generation Sequencing using Illumina MiSeq equipment (*Manso et al., 2020*). Sequences were trimmed of adaptors and low quality reads using Trimmomatic V. 0.33 (*Bolger et al., 2014*). The quality of the sequence files was then assessed using the FastQC program. The resulting FASTQ files were then aligned to the indexed reference J6/JFH HCVcc genome using the BWA-MEM algorithm (Burrows Wheeler Aligner *Li and Durbin, 2009*), converted into Sequence Alignment Map (SAM) files, which were further compressed into BAM files (binary versions of SAM files), sorted by reference coordinates and indexed using SAMtools. The duplicate sequences were then removed by Picard Tools and indexed again using SAMtools. Base-calling for each position in the genome was extracted from the indexed file. Positions of interest were identified as those with at least 1,000 reads available with variance in nucleotide base composition of ≥5%.

## Production of lentiviral vectors and receptor overexpression

To generate lentiviral vectors, HEK293T cells were transfected with three plasmids: an HIV packaging construct (pCMV-dR8.91), VSV-G envelope plasmid (pMD2.G) and a transfer plasmid encoding GFP and SR-B1 or CD81, expressed from separate promoters (pDual SR-B1 or CD81, available from Addgene: https://www.addgene.org/Joe_Grove/). Supernatants containing viral vectors were collected at 48 and 72 hr post-transfection. At least 96 hr before an experiment, Huh-7.5 were transduced with lentivirus vectors diluted in complete medium and 24 hr prior to study the cells were seeded into a 96 well plate for infection, as described above.

## Production of HCV pseudoparticles

To generate HCVpp, HEK293T^CD81KO cells (*Kalemera et al., 2021*) were co-transfected with three plasmids: an HIV packaging construct (pCMV-dR8.91), a luciferase reporter plasmid (CSLW) and an expression vector encoding the appropriate HCV glycoprotein. Supernatants containing HCVpp were collected at 48- and 72 hr post-transfection. Assays were carried out in an analogous manner to HCVcc, but with infection being measured after 48–72 hr by detection of luciferase expression using the BrightGlo assay kit (Promega). Chikungunya virus E1E2 expression plasmid was a gift from Brian Willett (University of Glasgow).

## Entry kinetics assay

Huh-7.5 cells were seeded for infection, as above. HCVpp were preincubated with magnetic nanoparticles (ViroMag, OZ Biosciences, France) for 15 min, added to Huh-7.5 cells and the plate was placed

on a Super Magnetic Plate (OZ Biosciences, France) for 15 min at 37 °C to synchronise infection (*Haim et al., 2009*; *Haim et al., 2005*). The synchronous infection was chased with a saturating receptor blockade by adding 3 µg/ml anti-CD81 mAb 2.131 at the indicated time points. Infection was assayed after 72 hr using the SteadyGlo reagent kit and a GloMax luminometer (Promega, USA).

## Production of sE2

Soluble E2 comprises residues 384–661 of the HCV genome, flanked by an N-terminal tissue plasminogen activator signal sequence and a C-terminal Twin-Strep-tag. HEK293T[CD81KO] cells (*Kalemera et al., 2021*) were transduced with lentivirus encoding sE2 and cell culture media, containing sE2, were harvested every 24 hours for up to 6 weeks. The harvested supernatants were frozen immediately at –80 °C. High purity monomeric sE2 was generated by sequential affinity purification using StrepTactin-XT columns (IBA Lifesciences, Göttingen, Germany) and size-exclusion chromatography using the HiPrep 16/60 Sephacryl S-200 HR gel filtration column (Cytiva, MA, USA). Soluble CD81 ectodomain (residues 113–201) was produced and purified in an analogous manner.

## Soluble E2 binding assay

The sE2 binding assay has been described elsewhere (*Kalemera et al., 2019*). Briefly, a single-cell suspension of $5 \times 10^5$ CHO cells transduced to express human SR-B1 or CD81 (protocol for receptor lentiviral production described above). Cells were preincubated in 'traffic stop' buffer, PBS + 1% bovine serum albumin and 0.01% sodium azide; the addition of $NaN_3$ depletes cellular ATP pools, consequently preventing active processes including receptor internalisation. All subsequent steps are performed in traffic stop buffer. Cells were pelleted and then resuspended in a serial dilution of sE2. Following a 1 hr incubation at 37 °C, cells were washed twice and incubated with 3 µg/ml StrepMAB-classic (for CHO SR-B1) or J6.36 (for CHO CD81) followed by an anti-mouse Alexa Fluor 647 secondary. After a final wash, the cells were fixed in 1% formaldehyde and analysed by flow cytometry. To measure cell surface expression of human receptors post-transduction, cells were stained for SR-B1 or CD81 and signal was detected using an anti-rabbit or anti-mouse Alexa Fluor 647 secondary, respectively.

## ELISA

Purified Strep-II-tagged sE2 monomers (1.0 µg/mL diluted in TRIS buffered saline, TBS) were added to 96-well Strep-TactinXT-coated plates (IBA Lifesciences, Göttingen, Germany) for 2 hours at room temperature. Plates were washed with TBS twice before incubating with serially diluted mAbs in casein blocking buffer (Thermo Fisher Scientific) for 90 min. After three washes with TBS, wells were incubated with a 1:3,000 dilution of HRP-labelled goat anti-human/mouse IgG in casein blocking buffer for 45 min. After washing the plates five times with TBS +0.05% Tween-20, plates were developed by adding develop solution (1% 3,3',5,5'-tetraethylbenzidine, 0.01% $H_2O_2$, 100 mM sodium acetate, 100 mM citric acid). The reaction was stopped after 3 min by adding 0.8 M $H_2SO_4$. Absorbance was measured at 450 nm.

## pH sensitivity assay

Experiment was adapted from *Sharma et al., 2011*. HCVpp were immobilised by incubation in a 96-well plate coated with poly-D-lysine (Thermo Fisher Scientific), unbound particles were rinsed away with PBS. HCVpp were then incubated for 15 min at 37 °C in neutral PBS, or PBS adjusted to pH 6 or pH 5. Following incubation, immobilised particles were rinsed in PBS, afterwhich $3 \times 10^4$ Huh-7.5 cells were added to each well. Experiments were read out as described above for HCVpp.

## Limited proteolysis

sE2 (0.2 mg/ml) was incubated with Endoproteinase GluC (New England Biosciences) at 1:50 (w/w) ratio in TBS at 37 °C for up to 4 hr. Proteolysis was halted by boiling in Laemmli buffer at specific intervals and the digestion products identified by SDS-PAGE followed by western blot.

## SDS-PAGE/western blotting

Samples were run on MiniPROTEAN 4–12% gels (BioRad, CA, USA) and transferred on to nitrocellulose membrane. The blots were blocked in PBS + 2% milk solution +0.1% Tween-20 and then probed

sequentially with J6.36 mAb and goat anti-mouse secondary conjugated to horseradish peroxidase. Chemiluminescence signal was then measured using a Chemidoc MP (BioRad, CA, USA).

## Circular dichroism spectroscopy

Circular dichroism experiments were performed using the B23 nitrogen-flushed Module B end-station spectrophotometer at B23 Synchrotron Radiation CD BeamLine located at Diamond Light Source. They were performed by the BeamLine lab manager as a mail-in service. Four scans of protein samples in 50 mM NaF and 20 mM $NaH_2PO_4$ and $Na_2HPO_4$ pH 7.2 were acquired in the far-UV region (175–260 nm) in 1 nm increments using an integration time of 1 sec, 1 nm bandwidth and pathlength of 0.00146 cm, at 20 °C. Results obtained were processed using CDApps v4.0 software (*Hussain et al., 2015*). The scans were averaged and spectra have been normalised using average amino acid molecular weight which was calculated for each sample. Spectra presented are difference spectra meaning the buffer baseline has been subtracted from the observed spectra, and zeroed between 253 and 258 nm. Secondary structure deconvolution from CD spectra was carried out using the CDApps CONTINLL algorithm and SP29 reference data set referencing 29 soluble protein structures.

## Nano differential scanning fluorimetry

The melting temperature ($T_m$) of sE2 (1 mg/ml) was calculated using the Prometheus NT.48 (Nano-temper, Munich, Germany) during heating in a linear thermal ramp (1 °C, 20–90°C) with an excitation power of 30%. The fluorescence at emission wavelengths of 350 and 330 nm was used to determine changes in tyrosine and tryptophan environments. The $T_m$ was calculated by fitting the Boltzmann sigmoidal curve to the first derivative of the fluorescence ratios (350 nm/330 nm).

## Mathematical modelling

We applied a mathematical model described in a previous publication (*Kalemera et al., 2019*) to the novel data generated by this study. Our model uses a series of differential equations to describe the processes via which HCV particles, having bound to the cell membrane, acquire CD81 and SR-B1 receptors. Viruses are modelled as uniformly having $N_e$ E2 proteins available to bind cellular receptors. We consider an E2 protein as being either bound or unbound to CD81. If it is unbound to CD81 we consider whether it is bound or unbound to SR-B1; once E2 is bound to CD81 we are unconcerned about its binding to SR-B1. Thus, we represent the population as existing on a grid of points $M_{ij}$, indicating viruses that have i copies of E2 bound to CD81 and j copies of E2 bound to SR-B1 but not to CD81. All viruses begin at the point $M_{00}$, then progress through the grid (see *Figure 5* in Kalemera et al. ).

At the point $M_{ij}$, viruses have $N_e$-i-j free E2. We suppose that they bind SR-B1 at some rate s (Figure S3). Viruses bind CD81 via two specific processes. Firstly, viruses bind CD81 via an SR-B1 independent process at the rate $c_1$. SR-B1 engagement provides a priming mechanism, which facilitates a second SR-B1-mediated acquisition of CD81, occurring at the rate $c_2$. Viruses which gain a pre-specified threshold number of CD81 receptors pass at some rate, e, into the downstream process of viral entry. Viruses die at constant rate d, which is arbitrarily pre-specified; in our model all other processes occur at some rate relative to this value. Viruses gaining entry to the cell are modelled as being in the state E, while viruses which are dead are modelled as being in the state D.

Our model considers data from a number of states in which the number of CD81 and SR-B1 receptors has been altered (*Figure 1C & D*). In a given system we model the proportion of CD81 receptors as pc, and the proportion of SR-B1 receptors as ps, where the value in unmodified cells is 1 in each case.

Following the above, we obtain the following equations for viral progress across the membrane:

$$\frac{dD}{dt} = d\sum_{ij} M_{ij}$$

$$\frac{dE}{dt} = e\sum_{j} M_{rj}$$

$$\frac{dM_{ij}}{dt} = \frac{1}{N_e}\left[p_c\left(c_1\left(N_e - i - j + 1\right)M_{i-1j} + c_2 j M_{i-1j+1}\right) + p_s s\left(N_e - i - j + 1\right)M_{ij-1}\right] - \ldots$$

$$\ldots \frac{1}{N_e}\left[\left(p_c\left(c_1\left(N_e - i - j\right)M_{i-1j} + c_2 j\right) + p_s s\left(N_e - i - j\right)\right)M_{ij}\right] - dM_{ij} - eI_{ir}M_{ij}$$

where $I_{ir}$ = 1 if i=r and $I_{ir}$ = 0 if i↑r, and the system has the initial conditions D=0, E=0, $M_{00}$=1, and $M_{ij}$ = 0 for all i>0 and j>0, and the system is bounded by the constraints 0 ″ i ″ r and 0 ″ j ″ $N_e$.

Given a set of input parameters, a fourth-order Runge-Kutte scheme with adaptive step size was used to propagate the system until the sum of terms D+E was greater than 0.999, indicating that 99.9% of the simulated viruses had either died or gained entry. The probability P that a single virus gains entry to a cell was then calculated as.

$$P\left(p_c, p_s, c_1, c_2, s, e\right) = \frac{E}{D+E}$$

An optimisation procedure was run to fit the model to data from experiments. As in a previous publication, experiments modelling viral entry were performed in replicate. To estimate the values $p_c$ and $p_s$, indicating the extent of available receptor, we used fluorescence microscopy measurements of receptor blockade/over-expression performed in parallel with the infection experiments, as previously described (**Kalemera et al., 2019**). These values were normalised to the range 0–1 in the case of receptor knockdown experiments, or above 1 in the case of overexpression experiments; 1 being the availability in unmodified cells.

We now consider data describing a level of receptor availability ($p_c$, $p_s$) and a number of observed foci of infection; using the index i we term the latter value $o_i$. Given a knowledge of the number of input particles and the number of cells observed in a well, this can be understood in terms of a probability of a given cell being infected by the virus. Observations were modelled as being distributed according to a double Poisson distribution with mean ſi and parameter ſ.

$$loglogL = loglog\left[\theta^{0.5}e^{-\theta\mu_i}\left(\frac{e^{-o_i}o_i^{o_i}}{o_i!}\right)\left(\frac{e\mu_i}{o_i}\right)^{\theta o_i}C\right]$$

where

$$\frac{1}{C} = 1 + \left(\frac{1-\theta}{12\theta\mu_i}\right)\left(1 + \frac{1}{\theta\mu_i}\right)$$

An estimate of the dispersion parameter θ was calculated by fitting a single value $\mu_i$ to each set of values $o_i$ arising from the same level of receptor availability with no other constraint on the $\mu_i$. This parameter was then used in the likelihood function to fit values $\mu_i = n_i P(p_c, p_s, c_1, c_2, s, e)$, where $n_i$ is the number of cells observed in a well and $P(p_c, p_s, c_1, c_2, s, e)$ is the probability of viral entry calculated from the differential equation model described above; parameters c1, c2, s, Ne, and e were optimised to give a maximum likelihood fit to the data.

## Molecular dynamic simulations

The complete model of the J6 E2 ectodomain was generated as previously described (**Stejskal et al., 2020**). For simulations of the mutant glycoprotein, the respective amino acid substitutions were modelled in using Modeller (**Webb and Sali, 2016**).

We performed MD simulations in explicit solvent using the Amber 16 GPU-based simulation engine (**Case et al., 2005**). The model was solvated in a truncated octahedral box using OPC water molecules. The minimal distance between the model and the box boundary was set to 12 Å with box volume of $4.2 \times 10^5$ Å$^3$. Simulations were performed using the ff14SB force field on GPUs using the CUDA version of PMEMD in Amber 16 with periodic boundary conditions. CONECT records were created using the in-house MakeConnects.py script to preserve the disulphide bonds throughout the simulations. MolProbity software was used to generate physiologically relevant protonation states.

Minimisation and equilibration: The systems were minimised by 1,000 steps of the steepest descent method followed by 9,000 steps of the conjugate gradients method. Sequential 1ns relaxation steps were performed using the Lagevin thermostat to increase the temperature from 0 to 310 K, with initial velocities being sampled from the Boltzmann distribution. During these steps, pressure was kept constant using the Berendsen barostat. All atoms except for the modelled residues, hydrogen atoms and water molecules were restrained by a force of 100 kcal/mol/Å$^2$. The restraint force was eventually decreased to 10 kcal/mol/Å$^2$ during subsequent 1ns equilibration steps at 310 K.

A further minimisation step was included with 1000 steps of the steepest descent method followed by 9,000 steps of the conjugate gradients method with all backbone atoms restrained by a force of 10 kcal/mol/Å$^2$. The systems were then subjected to four 1ns long equilibration steps at constant

pressure with stepwise 10-fold reduction of restraint force from 10 to 0 kcal/mol/Å². All minimisation and equilibration stages were performed with a 1fs time step.

Production runs: An initial 1μs production run was simulated under constant volume and temperature using the Langevin thermostat. SHAKE was used in all but the minimisation steps; this, in combination with hydrogen mass repartitioning, permitted 4 ft time-steps during the production runs. Short-range cutoff distance for van der Waals interactions was set to 10 Å. The long-distance electrostatics were calculated using the Particle Mesh Ewald Method. To avoid the overflow of coordinates, iwrap was set to 1. Default values were used for other modelling parameters. To achieve independent repeat simulations, we performed steps to decorrelate the output from the equilibration process. Initial velocities were generated from the Boltzmann distribution using a random seed. The coordinates, but not velocities, from the final equilibration step were used as input for a short (40ns) production run. The coordinates, but not velocities, from this run were used for a second 4ns production run. This was followed by a 1μs production run. This process was repeated for each independent simulation.

The MD trajectories were analysed using scripts available in cpptraj from Amber Tools 16. For RMSF/RMSD analyses, the average structure generated from the given trajectory was used as the reference structure. The analyses were performed using the backbone Cα, C and N atoms unless otherwise stated. MD trajectories are available here: https://zenodo.org/record/4309544.

## Molecular modelling

Molecular model visualisation was performed with UCSF Chimera (*Pettersen et al., 2004*).

## Statistical analysis

All statistical analysis was performed in Prism 6.0 (GraphPad, CA, USA). In the majority of cases ordinary one-way ANOVA was performed using Dunnett's multiple comparison test, using WT virus as a control. Unpaired t-test was performed assuming equal standard deviation using a two-tailed p-value. The F-test was used to compare fitted curves. Asterisks indicate level of significance: $p < 0.05$ = *, $p < 0.01$ = **, $p < 0.001$ = ***, $p < 0.0001$ = ****.

## Acknowledgements

We thank Prof. Greg Towers, Dr. Clare Jolly, Prof. Richard Milne, Dr. Clare Bennett and Prof. Ben Seddon for scientific and academic support, and to colleagues who provided resources (see Methods). We are grateful to Dr. Colman, Dr. Jackson and Dr. Louis for providing encouragement. For incisive criticism, we also thank Prof. Kermode, Prof. Mayo and HTJI. This work was supported by Sir Henry Dale Fellowships to J.G. (107653/Z/15/Z) and CJRI (101239/Z/13/Z); Medical Research Council CVR Core funding to JG (MC_UU_12014); a Wellcome Trust studentship to LS (109162/Z/15/Z); a Medical Research Council studentship to MDK (https://www.ukri.org/councils/mrc/); CJRI was also supported by Deutsche Forschungsgemeinschaft (DFG, Grant SFB 1310), the Medical Research Council (ref: MC_UU_00002/11) and the Isaac Newton Trust.

## Additional information

### Competing interests

Zisis Kozlakidis: Where authors are identified as personnel of the International Agency for Research on Cancer/WHO, the authors alone are responsible for the views expressed in this article and they do not necessarily represent the decisions, policy or views of the International Agency for Research on Cancer/WHO. The other authors declare that no competing interests exist.

### Funding

| Funder | Grant reference number | Author |
| --- | --- | --- |
| Wellcome Trust | 107653/Z/15/Z | Joe Grove |
| Medical Research Council | MC_UU_12014 | Joe Grove |

| Funder | Grant reference number | Author |
|---|---|---|
| Wellcome Trust | 101239/Z/13/Z | Christopher JR Illingworth |
| Deutsche Forschungsgemeinschaft | SFB 1310 | Christopher JR Illingworth |

The funders had no role in study design, data collection and interpretation, or the decision to submit the work for publication. For the purpose of Open Access, the authors have applied a CC BY public copyright license to any Author Accepted Manuscript version arising from this submission.

## Author contributions

Lenka Stejskal, Tina Daviter, Formal analysis, Investigation, Methodology; Mphatso D Kalemera, Formal analysis, Investigation, Methodology, Writing – review and editing; Charlotte B Lewis, Machaela Palor, Lucas Walker, Myrto Kremyda-Vlachou, Giulia Gallo, Investigation; William D Lees, David S Moss, Methodology; Zisis Kozlakidis, Supervision, Methodology; Dalan Bailey, Supervision; William Rosenberg, Resources, Methodology; Christopher JR Illingworth, Formal analysis, Investigation, Methodology, Funding acquisition; Adrian J Shepherd, Formal analysis, Supervision, Funding acquisition, Methodology; Joe Grove, Conceptualization, Resources, Data curation, Formal analysis, Supervision, Funding acquisition, Investigation, Methodology, Writing – original draft, Project administration, Writing – review and editing

## Author ORCIDs

Mphatso D Kalemera http://orcid.org/0000-0001-9461-1117
Dalan Bailey http://orcid.org/0000-0002-5640-2266
William Rosenberg http://orcid.org/0000-0002-2732-2304
Christopher JR Illingworth http://orcid.org/0000-0002-0030-2784
Adrian J Shepherd http://orcid.org/0000-0003-0194-8613
Joe Grove http://orcid.org/0000-0001-5390-7579

## Ethics

Human subjects: Fully consented blood samples (for IgG isolation) were collected from HCV+ patients under ethical approval: "Characterising and modifying immune responses in chronic viral hepatitis"; IRAS Number 43993; REC number 11/LO/0421.

## Decision letter and Author response

Decision letter https://doi.org/10.7554/eLife.71854.sa1
Author response https://doi.org/10.7554/eLife.71854.sa2

# Additional files

## Supplementary files
• Transparent reporting form
• Source data 1. All underlying data from each experimental measurement is included. Large data sets from molecular dynamic experiments are excluded, however, full MD trajectories are available for download.

## Data availability
The underlying data for this manuscript are provided as a Source Data file. Full molecular dynamic simulation trajectories are available here: https://zenodo.org/record/4309544.

The following dataset was generated:

| Author(s) | Year | Dataset title | Dataset URL | Database and Identifier |
|---|---|---|---|---|
| Shepherd AJ, Stejskal L, Grove J | 2020 | MD simulation data: An Entropic Safety Catch Controls Hepatitis C Virus Entry and Antibody Resistance | https://zenodo.org/record/4309544#.YuAAbezMIUE | Zenodo, 10.5281/zenodo.4309544 |

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
