## [Editor Report]

HCV is unique in its glycoprotein structure, complex receptor usage and an unusual persistence for a (+)RNA virus. In this well done study, the authors explain a number of observations regarding receptor usage and how HCV evades antibody control via HVR1, whose disordered nature enable mutation to continually evade antibody responses.

---

## [Decision Letter]

**Decision letter after peer review:**

Thank you for submitting your article "An Entropic Safety Catch Controls Hepatitis C Virus Entry and Antibody Resistance" for consideration by *eLife*. Your article has been reviewed by 3 peer reviewers, one of whom is a member of our Board of Reviewing Editors, and the evaluation has been overseen by Vivek Malhotra as the Senior Editor. The reviewers have opted to remain anonymous.

Essential revisions:

Lines 356-358: In other genotypes, I438 can be an L or V, while A524 has been observed as V, K, N, or I; other substitutions could also occur. If the infectivity/stability trade-off model is a feature of the natural history of HCV as a chronic infection, then each of these genotypes would be expected to fall along a spectrum of that trade-off and shift when the double variant is introduced. In order to make this claim about changing selection pressures, please test whether the infectivity/stability trade-off model holds true in other genotypes.

If the mechanism of this difference is premature progression into the post-fusion state, then fusion assays will reflect this. There are several assays available (see Denolly et al., Membrane fusion assays for studying entry hepatitis C virus into cells, 2019). Additionally, if the double mutant E2 is irreversibly progressing to the post-fusion state, then the molecular dynamics model should reflect that much lower energy state and yet is unchanged. Also on this theme, VSV-G is reversible so not all post-fusion states are irreversible (line 43).

To compare the stability/infectivity of wild type/mutant virions upon low pH treatment. Also, one can acid-prime HCV to enter at the cell surface – does this result in enhanced entry for the mutants? This might help relate changes in E2 to E1 fusion activity?

*Reviewer #2 (Recommendations for the authors):*

Lines 356-358: In other genotypes, I438 can be an L or V, while A524 has been observed as V, K, N, or I; other substitutions could also occur. If the infectivity/stability trade-off model is a feature of the natural history of HCV as a chronic infection, then each of these genotypes would be expected to fall along a spectrum of that trade-off and shift when the double variant is introduced. In order to make this claim about changing selection pressures, please test whether the infectivity/stability trade-off model holds true in other genotypes.

If SR-BI is enhancing but not essential according to the model described lines 152-157, then the increased titers in the variant (figure 1A) could explain compensation for SR-BI knock out (figure 1B). Titers should either be controlled or normalized to clarify this point.

Line 309: allostery, here, is conjecture. Particularly when the model shows an absence of information on this point.

Line 374: Typically, allostery requires a higher burden of proof than direct action. No solid evidence of allostery is present in figure 5.

If the mechanism of this difference is premature progression into the post-fusion state, then fusion assays will reflect this. There are several assays available (see Denolly et al., Membrane fusion assays for studying entry hepatitis C virus into cells, 2019). Additionally, if the double mutant E2 is irreversibly progressing to the post-fusion state, then the molecular dynamics model should reflect that much lower energy state and yet is unchanged. Also on this theme, VSV-G is reversible so not all post-fusion states are irreversible (line 43).

Figure 1C and D values are highly variable in 4 replicates, generating a pretty but not particularly meaningful curve. Since the mathematical model in figure 2 is an extrapolation of this data, perhaps a residuals or other quality-of-fit metric is needed.

Figure 4 D, consider giving at least three independent superimposed frames from MD run at a different time interval (early, middle, and late). Average values could be biased.

Limited proteolysis with Glu C was used to evaluate the subtle structural differences of I438V A524T compared to WT. What’s the rationale behind choosing GluC over other digestive enzymes? Also, quantification of two different Western blots is not as accurate. Should consider using less time points on the same gel.

Do you have any explanation how I438V A524T mutation modulating SR-B1 binding since these mutations are not in or close to HVR1 region of E2?

*Reviewer #3 (Recommendations for the authors):*

The manuscript is very nicely written and makes a complex concept nicely acceptable to a general interest audience.

I would suggest the following experiments/investigations/changes might improve the manuscript:

1. To compare the stability/infectivity of wild type/mutant virions upon low pH treatment. Also, one can acid-prime HCV to enter at the cell surface – does this result in enhanced entry for the mutants? This might help relate changes in E2 to E1 fusion activity?

2. MD – could the diagrams be made a little larger and with some extra annotation that signifies any regions of interest seen in during the simulations that differ between wild type and mutant? Does HVR1 interact with other parts of E2 that are then left unbound when the region becomes stabilised?

3. Are there examples of non-neutralising abs targeting HVR1? Could Fabs be used to induce the safety catch if so?

4. Was sequencing carried out in the antibody selected cultures? If not, then an explanation as to why is necessary, or if data exist they should be included.

---

## [Author Response]

Reviewer #2 (Recommendations for the authors):Lines 356-358: In other genotypes, I438 can be an L or V, while A524 has been observed as V, K, N, or I; other substitutions could also occur. If the infectivity/stability trade-off model is a feature of the natural history of HCV as a chronic infection, then each of these genotypes would be expected to fall along a spectrum of that trade-off and shift when the double variant is introduced. In order to make this claim about changing selection pressures, please test whether the infectivity/stability trade-off model holds true in other genotypes.

We agree that a weakness of our original work was that it only utilised the J6 clone E1E2 (genotype 2a). Therefore, we introduced the analogous L438V A524T hyper-reactive mutations in to the H77 clone (genotype 1a) and phenotyped the resultant E1E2 using the HCVpp system, paired with molecular dynamics simulation of E2. This new data is included in a revised Figure 6 (and lines 382-393) and is supportive of our central hypothesis. H77 L438V A524T E1E2 bears the hallmarks of a hyper-reactive virus, exhibiting increased sensitivity to antibodies, decreased dependence on SR-B1 and reduced dynamics in HVR-1. This suggests that the entropic-safety catch model is applicable to other genotypes/clones of HCV.

If SR-BI is enhancing but not essential according to the model described lines 152-157, then the increased titers in the variant (figure 1A) could explain compensation for SR-BI knock out (figure 1B). Titers should either be controlled or normalized to clarify this point.

The data normalisation in both of these figures already accounts for this possibility. In Figure 1A, the virus particle input is normalised by genome copy number. In Figure 1B, the data is normalised to infection of unmodified parental Huh-7 cells for each respective virus and, therefore, displays receptor dependency independent of absolute titre.

Line 309: allostery, here, is conjecture. Particularly when the model shows an absence of information on this point.Line 374: Typically, allostery requires a higher burden of proof than direct action. No solid evidence of allostery is present in figure 5.

We agree that our interpretation here went further than our data allowed. We have removed the first statement around allostery and significantly weakened the second statement (line 424) such that allostery is only considered as a possible explanation.

If the mechanism of this difference is premature progression into the post-fusion state, then fusion assays will reflect this. There are several assays available (see Denolly et al., Membrane fusion assays for studying entry hepatitis C virus into cells, 2019).

We agreed that a fusion assay would enhance our investigation and therefore set up experiments using Huh-7 cells expressing a split-GFP/luciferase reporter system. We were able to consistently measure cell-cell fusion mediated by H77 E1E2, however, despite extensive optimisation we were unable to measure fusion by clone J6 or J6 I438V A524T. Optimisation steps included testing of buffer constituents and pH, alterations in seeding density, changing donor and acceptor cell-types and codon optimisation of the J6 E1E2 expression plasmids. We include in Author response image 1, an example comparing cell-cell fusion by H77 E1E2, empty pD603 control plasmid the respective J6 E1E2 plasmids with and without codon optimisation. Furthermore, H77 L438V A524T E1E2 (Figure 6) was also unable to mediate fusion, likely due to reductions in E1E2 stability.

**Author response image 1. sa2fig1:** HCV E1E2 fusion assay. Two populations of Huh-7 cells expressing matched components of the split-GFP/luciferase system were co-seeded and transfected with the stated E1E2 expression plasmid (or pD603 empty plasmid). 48 hours later, cell fusion was triggered by 10 minute incubation in PBS adjusted to pH 5. Fusion was assessed after 24 hours by the measurement of reconstituted luciferase signal. CO = codon optimised plasmids.

Nonetheless, our additional, successful, experiments examining E1E2 pH sensitivity (see below) do inform on the fusogenic propensity of the hyperreactive mutant. Moreover, we have weakened and nuanced the text to ensure we do not inappropriately interpret experiments in respect to fusogenicity. We, therefore, believe that the statements, interpretations and hypotheses made in our revised manuscript are well supported by data, and we have adequately addressed the nature of this comment.

Additionally, if the double mutant E2 is irreversibly progressing to the post-fusion state, then the molecular dynamics model should reflect that much lower energy state and yet is unchanged.

In respect to this point, membrane fusion by HCV requires the concerted action of both E2 and, the minor glycoprotein, E1. Since the structure of this functional complex remains unknown, our MD experiments only examine E2. Consequently, we would not expect our study to capture the conformational changes, and energetic transitions, that are required for fusion. Indeed, we do not interpret our MD experiments in respect to fusogenic change.

Also on this theme, VSV-G is reversible so not all post-fusion states are irreversible (line 43).

We have corrected this error, whilst also reflecting that most viral fusion proteins undergo irreversible conformational change (line 49).

Figure 1C and D values are highly variable in 4 replicates, generating a pretty but not particularly meaningful curve. Since the mathematical model in figure 2 is an extrapolation of this data, perhaps a residuals or other quality-of-fit metric is needed.

Importantly, Figure 1C and D display normalised data that demonstrate the relative inhibition of each virus, whereas our modelling was performed on the underlying counts of infected cells. Nonetheless, we appreciate the nature of the reviewer’s concern. To address this we have revised Figure 2-Supplement 1 to include plots that directly demonstrate the relationship between the underlying data points and the fits generated by the mathematical model.

Figure 4 D, consider giving at least three independent superimposed frames from MD run at a different time interval (early, middle, and late). Average values could be biased.

Figure 4D represents static models which we have colour-coded for flexibility, as measured across five independent MD simulations. The type of representative superimposed frames that the reviewer is requesting are found in Figure 4A, Figure 4 Supplements 1 and 2, Figure 6E, and Figure 6 Supplement 1.

Limited proteolysis with Glu C was used to evaluate the subtle structural differences of I438V A524T compared to WT. What's the rationale behind choosing GluC over other digestive enzymes?

Prediction of protease cleavage sites within E2 followed by empirical testing of various candidates guided our choice of enzyme. In particular, GluC generated a digest fragment that could be consistently detected by Western blotting and with kinetics that we could reliably capture in our experimental setup.

Also, quantification of two different Western blots is not as accurate. Should consider using less time points on the same gel.

To ensure cross comparability of these experiments, the limited proteolysis densitometry data were normalised to an internal control (band intensity after 4 hours incubation). This is analogous to the internally normalised antibody neutralisation curves in Figure 3. Consequently, the data allow direct comparison of the kinetics of digestion, independent of inter-gel/blot variation.

Therefore, we do not believe that running additional Western blots will improve the quality of our data.

Do you have any explanation how I438V A524T mutation modulating SR-B1 binding since these mutations are not in or close to HVR1 region of E2?

We did not identify consistent interactions or structural repositioning that could easily explain the changes in HVR-1 dynamics; this forms part of our ongoing investigation. We have added a statement to the text making this clear (line 426).

Reviewer #3 (Recommendations for the authors):The manuscript is very nicely written and makes a complex concept nicely acceptable to a general interest audience.I would suggest the following experiments/investigations/changes might improve the manuscript:1. To compare the stability/infectivity of wild type/mutant virions upon low pH treatment. Also, one can acid-prime HCV to enter at the cell surface – does this result in enhanced entry for the mutants? This might help relate changes in E2 to E1 fusion activity?

We performed additional experiments to evaluate the pH sensitivity of E1E2

(Figure 3 Supplement 3 and lines 239-257). These proved very informative and allow a clearer understanding of how the safety catch model relates to pH sensing and fusion. The key findings are: (i) the hyper-reactive mutant has no change in its intrinsic pH sensitivity (like WT, it is completely resistant to acidic buffers) but (ii) it is more prone to CD81-mediated priming of pH sensing. This is consistent with a step-wise priming of E1E2 fusion, with toggling of the HVR-1 safety catch being necessary, but not sufficient to activate pH sensing/fusion.

2. MD – could the diagrams be made a little larger and with some extra annotation that signifies any regions of interest seen in during the simulations that differ between wild type and mutant?

HVR-1 is the only region that significantly differs between WT E2 and the various mutants. We have however, amended the Figures 4 and 6 to emphasise the colour coded, superimposed, images of HVR-1 mobility.

Does HVR1 interact with other parts of E2 that are then left unbound when the region becomes stabilised?

We were unable to identify consistent interactions between HVR-1 and other regions of E2. We have included a statement to make this clear (line 426).

3. Are there examples of non-neutralising abs targeting HVR1? Could Fabs be used to induce the safety catch if so?

We had already performed experiments to test this idea and initial data were encouraging, with anti-HVR-1 enhancing interaction between E2 and CD81. However, it has been difficult to conclusively demonstrate that this is due to HVR-1 ligation and not avidity effects due to crosslinking of adjacent E2 molecules by bivalent antibodies. In particular, fAb fragments demonstrated very weak binding to E2 and minimal enhancement of CD81 interaction. Further iterations of this experiment will form the basis of on going investigations.

4. Was sequencing carried out in the antibody selected cultures? If not, then an explanation as to why is necessary, or if data exist they should be included.

In the course of on going work we performed further analyses of later timepoints from the experiment originally presented in Figure 6. This proved much more conclusive regarding the emergence of hyper-reactive mutants in the absence of nAb selection. The revised Figure 6, and text (lines 369-380), includes a summary of sequencing data from the cultures with and without IgG selection and characterisation of an additional hyper-reactive mutant virus.